# 🎯BOW: Reinforcement Learning for Bottle-necked Next-Word Prediction

## Abstract

Large language models (LLMs) are typically pretrained with next-word prediction (NWP), which yields strong surface fluency but places limited pressure on models to form explicit reasoning before emitting tokens. We study whether shifting the supervision signal can better elicit explicit reasoning and, more broadly, strengthen models' general reasoning capability. We present BOttlenecked next-Word exploration (BOW), a RL formulation of NWP that inserts an intermediate reasoning bottleneck. Instead of predicting the next word directly from context, the policy model must first generate a next-word reasoning trajectory. A frozen scorer then assigns this trajectory a soft, distributional reward equal to the probability of the gold next token conditioned solely on the trajectory to guide the RL optimization. We also propose an optional L1-style regularizer on the reward to discourage "name-the-answer" shortcuts. Across ten benchmarks, a brief BOW adaptation phase on Qwen2.5-7B-Instruct and Llama3.1-8B-Instruct improves zero-shot reasoning and outperforms strong continual-pretraining baselines, including an RL variant with a hard, binary reward and a supervised finetuning approach with augmented data, by nearly 5% on average, while achieving the top result in 7 of 10 intrinsic NWP evaluations. These results indicate that BOW is a viable alternative to vanilla NWP, inducing explicit next-word reasoning and strengthening general reasoning ability.

## 1 Introduction

Large language models (LLMs) have achieved remarkable progress in a broad range of natural language understanding and generation tasks, largely powered by the next-word prediction (NWP) objective during pretraining and supervised finetuning during post-training (Devlin et al., 2019; Raffel et al., 2020; Brown et al., 2020; Achiam et al., 2023). NWP is widely regarded as a principal mechanism through which LLMs acquire factual world knowledge and, to a significant extent, reasoning and decision-making capabilities (ichter et al., 2023; Yao et al., 2023; Shah et al., 2025; Ruis et al., 2025). However, recent works reveal fundamental limitations in standard NWP: always providing the gold next word as supervision can both obscure rich underlying information and encourage models to latch onto superficial correlations in the data, ultimately resulting in brittle generalization and hallucinations (Zhou et al., 2024; Li et al., 2024; Xu et al., 2025); moreover, as recent theory argues, prevailing pretraining and evaluation practices tend to reward confident guessing over calibrated uncertainty, further entrenching such failures (Kalai et al., 2025). Essentially, the standard objective in NWP supervises what comes next more than *why* it should follow, leaving little pressure to construct explicit, verifiable reasoning before predicting the next token. Prior works (Jiang et al., 2024; Xu et al., 2025; Ishibashi et al., 2025) have demonstrated that explicitly encouraging models to articulate why certain continuations are likely leads to substantial gains in reasoning and reliability.

In this work, we take this line of inquiry further and introduce BOttlenecked Next Word Exploration (BOW), a RL framework that fundamentally rethinks the NWP task. Rather than directly supervising the policy

model to predict ground-truth next words, we instead bottleneck the learning process. The policy model must first explore to generate comprehensive and self-contained next-word reasoning trajectories, without seeing the actual next word. Subsequently, an off-the-shelf frozen scorer model evaluates the quality of the next-word reasoning trajectory with respect to the ground-truth next token, producing a soft, distributional reward signal for policy optimization. This signal quantifies the extent to which the ground-truth token is recoverable from the trajectory alone. By replacing the conventional NWP objective with BOW, we compel models to move beyond surface-level correlations and explicitly construct reasoning trajectories carrying sufficient information to recover plausible next words. We further introduce an optional reward-regularization term that prevents trajectories from directly over-narrowly verbalizing one or a few candidates ("name-the-answer" behavior), but encourages them to integrate comprehensive contextual cues to reason and characterize the space of plausible next words.

Concurrently, Dong et al. (2025) proposes Reinforcement Pre-Training (RPT), which also reframes NWP under RL but uses a hard, binary reward: trajectories are rewarded only if the final predicted token exactly matches the ground truth. However, the binary signal is counterintuitive since many words are unpredictable in natural language. Optimizing a model with a binary reward is more likely to induce unwanted behavior, such as hallucination, as demonstrated later in experiments on the TruthfulQA benchmark. In contrast, BOW's soft, distributional reward design assigns partial credit in proportion to how strongly a reasoning trajectory increases the probability of the ground next token to appear beyond hard-match, directly shaping how the trajectory supports multiple plausible continuations rather than only rewarding exact hits. Our subsequent experiments demonstrate the superior performance of BOW. Moreover, we provide an informal theoretical analysis from an information theory perspective in §A to demonstrate that our soft reward provides more information per sample, resulting in improved exploration efficiency.

Across ten benchmarks spanning world knowledge, multi-hop reasoning, and factual reasoning, a brief BOW adaptation phase on Qwen2.5-7B-Instruct and Llama3.1-8B-Instruct yields superior zero-shot general-reasoning performance, surpassing various strong baselines utilizing the same set of supervision tokens, including RPT by $\sim 5\%$ on average. For an intrinsic evaluation, we transform existing benchmarks into next-word prediction format, and demonstrate that BOW attains the top performance in 7 out of 10 evaluation settings compared with strong baselines, including RPT and Thought of Words (ToW) (Xu et al., 2025). Our ablation studies further validate the robustness of BOW to the choice of the frozen scorer model and the effectiveness of reasoning-centric training data filtering, a token selection process we apply before RL. Finally, human analyses indicate that BOW yields the most comprehensive next-word reasoning traces with general descriptions of plausible next words, whereas the hard, binary reward leads to the least satisfaction. At the same time, BOW without reward regularization yields the most human-aligned next-word prediction reasoning, as indicated by the human study. Taken together, these results support BOW as a scalable alternative to vanilla NWP to induce explicit next-word reasoning and strengthen general reasoning ability.

## 2 RELATED WORK

**Elaborated Reasoning.** Recent research has increasingly emphasized the importance of encouraging LLMs to articulate their internal reasoning processes rather than directly emitting final answers. ToW (Xu et al., 2025) demonstrates that continually pretraining models to generate reasoning before producing the next word improves factuality and general reasoning across a range of tasks. Similarly, Jiang et al. (2024) explores training models to produce natural language rationales between sentences for downstream reasoning tasks, showing that explicit reasoning improves both answer quality and user trust. Moreover, Ishibashi et al. (2025) proposes an unsupervised method to uncover and amplify implicit reasoning signals in domain-specific corpora, mimicking the human thinking processes involved in creating texts. These works motivate our design of a bottlenecked reasoning step that compels models to externalize their thought processes before prediction in a self-evolving way through reinforcement learning.

**Bottlenecked Learning.** Recent work explores forcing models through information bottlenecks–constrained intermediate representations that must capture essential reasoning before prediction. Zelikman et al. (2022) introduces Self-Taught Reasoner (STaR), which creates a reasoning bottleneck by requiring models to generate explicit rationales before answers. The model can only access the final answer through this rationale bottleneck, iteratively learning which reasoning paths lead to correct predictions. Zelikman et al. (2024) further extends this to Quiet-STaR, creating an even tighter bottleneck where models must generate "thoughts" with a fixed length between every token during pretraining, not just for explicit questions. More recently, Zhou et al. (2025) demonstrates that bottlenecking can operate at an even more abstract level—forcing models to recognize and transfer high-level reasoning patterns rather than surface-level associations, creating a conceptual bottleneck that enables generalization to rare cases. Our BOW framework implements a particularly stringent form of bottlenecked learning: without ever seeing the gold token directly, the policy model must generate reasoning that successfully restores next token prediction, rewarded with a soft, distributional signal. This architectural bottleneck ensures that the reasoning must contain sufficient information for an independent frozen scorer to recover the correct prediction. In contrast, RPT (Dong et al., 2025) optimizes language models with a binary reward, which renders positive feedback extremely sparse in early training, particularly in settings with large vocabularies, ambiguous contexts, or an initially weak policy model (notably, RPT is initialized from R1-distilled Qwen models). Thus, their optimization is driven by infrequent "jackpot" hits, which yield high-variance gradients and unstable exploration. However, our reward is dense and smooth: trajectories that marshal partial but relevant evidence still earn proportional credit, thereby stabilizing exploration.

**Reasoning Overfitting.** A growing body of work reveals that LLMs often exploit spurious correlations rather than performing genuine reasoning, a phenomenon sometimes termed as reasoning overfitting. Li et al. (2024) shows that LLMs take deceptive semantic shortcuts, relying on keyword/entity biases instead of following correct reasoning chains. This aligns with findings from Zhou et al. (2024), which demonstrates that LLMs drop 9-28% in performance when forced to reason abstractly rather than relying on surface patterns. In mathematics, which requires much rigor in reasoning, Yu et al. (2024) and Li et al. (2025) have both shown that LLMs' so-called math reasoning primarily relies on pattern matching and memorization of solution paths from training data, often establishing spurious correlations between surface-level features and certain mathematical concepts. These works collectively highlight that current LLMs often produce plausible-looking reasoning that masks fundamental failures in logical coherence. Our bottlenecked reasoning approach BOW addresses this by requiring models to generate reasoning that must successfully guide a frozen next-token scorer, providing an external validation of reasoning quality beyond surface plausibility.

## 3 BOTTLENECKED NEXT WORD EXPLORATION

### 3.1 OVERVIEW

We formulate our Bottlenecked Next Word Exploration problem as a reinforcement learning framework, outlined in Fig. 1. The policy model samples next-word reasoning trajectories through a bottlenecked generation process: instead of directly conditioning on the preceding context for next-word prediction, the policy model generates intermediate reasoning trajectories to infer plausible next-word candidates. The reward for each reasoning trajectory is computed in a self-verifiable manner by a frozen scorer, depending solely on the trajectory content and the ground next-word. The policy model is then optimized using GRPO (Shao et al., 2024) to maximize expected trajectory rewards. This formulation decouples next-word prediction from direct context conditioning, introducing an explicit reasoning bottleneck that enhances interpretability while maintaining generation quality through reward-based optimization.

**Context**

I like fruits, so for my lunch. I ate two

**Policy Model**

$r(\tau)$ = **0.5**

**Probs**

| apple | 0.50 |
| grape | 0.02 |
| broccoli | 0.01 |

**Frozen Scorer**

NWP

**Reasoning Trajectory $\tau$ for Next Word**

The next word set will likely contain nouns that are fruits, and these fruits should be larger in size to fill up for lunch. The reasoning is as follows:

Given the context "I like fruits a lot, so for lunch, I ate two," the next word set will likely contain nouns that are fruits, and these fruits should be larger in size to fill up for lunch. The reasoning is as follows:

1. The phrase "I like fruits a lot" indicates a preference for fruits, narrowing down the possible next words to fruits.
2. The phrase "for lunch" suggests that the next word should be a food item suitable for a meal.
3. The phrase "I ate two" implies that the fruits are in pairs, which are likely to be larger fruits that can be consumed in pairs, such as apples, pears, or bananas.

Therefore, the next word set can be described as: **fruits that can be eaten in pairs for lunch**.

Figure 1: **An overview of BOW**. Given the context, "*I like fruits a lot, so for my lunch, I ate two*" and its ground-truth next word, "*apple*", rather than predicting "*apple*" directly, the policy model first samples intermediate next-word reasoning trajectories. Then, a frozen LLM scorer (❄) is prompted with, "*... Given solely this context, the immediate next word is*", to obtain a next-token probability distribution. Finally, we formulate the reward of the reasoning trajectory as the probability of the ground-truth next token in this distribution. The policy model is finally optimized with GRPO.

## 3.2 BOTTLENECKED GENERATION

In traditional NWP, models are trained to predict the next word given a preceding context $C$. In contrast, BOW introduces a structural bottleneck: rather than directly predicting the next word, the policy language model $\pi_\theta$ must first generate an intermediate reasoning trajectory $\tau$ to reason towards plausible next words without directly seeing the gold next word $w^*$. This bottleneck process fundamentally changes the learning schema from a one-step classification task to a multi-step generative decision-making process, where the reasoning path $\tau$ serves as a latent action. Notably, the gold next word $w^*$ is never observed by the policy model nor used in any loss calculation. Supervision is provided only through a scalar reward signal assessing the informativeness and correctness of the generated reasoning path given $w^*$.

To ensure the learning remains feasible in our under-specified setting, we carefully design the policy model prompt to elicit reasoning trajectories that exhibit two critical properties: comprehensively incorporating all relevant contextual features that influence next word reasoning, and providing a general characterization of plausible next words rather than explicitly identifying specific candidate words. For example, in Fig. 1, pink texts reflect the first property and the blue texts reflect the second, given the context. A solid prompting design provides a strong starting point for generating reasoning trajectories and has been shown effective in prior work (Gandhi et al., 2025) to facilitate reasoning supervision in similar low-resource or weakly supervised regimes. Please refer to Fig. 3 in the Appendix for concrete prompt design.

## 3.3 REWARD RECIPE

To assess the quality of a next-word reasoning trajectory $\tau$, we leverage a frozen instruction-tuned language model $J_\phi$ as the trajectory scorer. We prompt $J_\phi$ with a simple instruction to predict the most likely next word conditioned on the reasoning trajectory $\tau$. We use $P_{J_\phi}(\cdot|\tau)$ to represent the immediate next token probability distribution conditioned on $\tau$, given by $J_\phi$. Thus, for any possible next token $w$, its probability to be inferred as the true next token is $P_{J_\phi}(w|\tau)$. Finally, we define the reward for the reasoning trajectory $\tau$ as the conditional probability of the true next token $w^*$ under $P_{J_\phi}(\cdot|\tau)$:

$$r(\tau) := P_{J_\phi}(w^* \mid \tau) \tag{1}$$

Notice that $J_\phi$ is never post-trained to perform such tasks, such as scoring completions step-by-step (Lightman et al., 2024); instead, it only performs a constrained continuation task as an off-the-shelf model: predicting the next token given a structured intermediate rationale. This design explicitly creates an information bottleneck and implicitly evaluates the generated reasoning trajectory by the effectiveness of recovering next tokens, encouraging a self-contained and comprehensive next-word reasoning from the policy model. To justify that $P_{J_\phi}(\cdot|\tau)$ is capable of reliably approximating the likelihood of candidate next words as described in the next-word reasoning trajectory, we provide a concrete example in Fig. 2 in the Appendix.

Although we later demonstrate that optimization under the current reward design yields strong performance across benchmarks, our human analysis reveals occasional over-narrowing of the trajectory, where the model explicitly enumerates only a few next-word candidates rather than leveraging contextual cues to perform in-depth next-word reasoning and provide general next-word descriptions. Such behavior is particularly undesirable in high-stakes domains, such as medical and finance, where over-narrowing reasoning can potentially lead to safety concerns, overcommitment, and hallucination (Kalai et al., 2025). We provide such concrete examples to demonstrate in §D. To address this issue, we introduce an additional L1-style regularization term to the reward, motivated by the observation that widely adopted instruction-tuned models have already internalized useful patterns and behaviors (Gandhi et al., 2025) when reasoning over plausible next words given a context. Specifically, we obtain a reference next-token distribution $P_{J_\phi}(\cdot|C)$ by directly prompting $J_\phi$ to provide a next word given the context $C$, restricting the optimization from excessive deviation. The final reward with regularization is formulated below, where $\alpha$ represents a scaling factor.

$$R(\tau) = r(\tau) - \alpha \| P_{J_\phi}(\cdot|\tau) - P_{J_\phi}(\cdot|C) \|_1 \tag{2}$$

### 3.4 POLICY OPTIMIZATION

We optimize the policy model $\pi_\theta$ with Grouped Reward Policy Optimization (GRPO) (Shao et al., 2024), which improves training stability by normalizing rewards within groups of reasoning paths sharing the same context. For each context $C$, a group of $N$ reasoning trajectories $\{\tau_1, \ldots, \tau_N\}$ are sampled from $\pi_\theta$. Reward $R(\tau_i)$ for each reasoning path $\tau_i$ is computed using Eq. 2. GRPO uses the group mean $\bar{r}$ and group standard deviation $\sigma$ to compute advantages $\hat{A}_i = \frac{r_i - \bar{r}}{\sigma}$ for each $\tau_i$, reducing gradient variance. Policy model updates are then performed using PPO-style optimization (Schulman et al., 2017). In fact, any RL algorithms can be applied in BOW, such as PPO (Schulman et al., 2017) and REINFORCE (Williams, 1992).

## 4 EXPERIMENTS

**Training Data.** We train our models on narratives from the murder mystery domain (Del & Fishel, 2023), which we argue is well-suited for studying reasoning-driven next-word prediction. Mystery stories naturally encode complex world models—they describe who did what, when, why, and what happened next—requiring both commonsense and counterfactual reasoning to interpret. In this sense, we view next-word prediction not just as a language modeling task, but as an implicit approximation of world state transitions. Story-driven data thus provides rich, structured input–output sequences that align well with our goal of encouraging explicit reasoning in LLMs. Concretely, we use 191 long-form narratives [1] sourced from the "5 Minute Mystery" platform,[2] and then filter out those that exceed 2048 tokens, yielding 178 narratives for training. To focus learning on reasoning-relevant supervision signals, we further filter the training data to remove context–next word pairs where the next tokens do not require meaningful reasoning to derive based on the context. The filtering process retains a final dataset of approximately 45K context–next word pairs, with more details in §B.1. By aligning training examples with tokens that genuinely demand reasoning, we ensure that the supervision signal is compatible with the bottlenecked learning setup of BOW, where the model is rewarded not for token overlap, but for the quality of its latent reasoning.

---

[1] https://github.com/TartuNLP/true-detective
[2] https://www.5minutemystery.com/

**RL Optimization.** We use Qwen2.5-7B-Instruct (Team, 2024) and Llama3.1-8B-Instruct (Grattafiori et al., 2024) as policy models (we will use Qwen2.5-7B-I and LLaMA3.1-8B-I to represent for short). We train one epoch with a total batch size of 1024, a mini-batch size of 256, and a rollout size of 5. We turn off the KL loss used in standard GRPO. We employ LLaMA3.1-8B-I as $J_\phi$ in §3.3 to calculate rewards for both policy model variants. Justification for such a scorer choice is provided in §C, but we also show BOW is robust to various scorer choices in the ablation study §6. To calculate $P_{J_\phi}(\cdot \mid \tau)$ in Eq. 1, we use a temperature of 5 for a smooth numerical distribution. When the first token of the gold next word appears among the top 100 tokens with the highest probabilities, the corresponding trajectory is assigned a reward as in Eq. 1, otherwise, 0. For the optional regularization calculation in Eq. 2, we use the top 100 tokens with the highest probabilities from $J_\phi(\cdot \mid C)$. We set $\alpha$ to 0.1 to roughly match the numerical scale of the regularization term with the initial reward. Please refer to §B.3 for more implementation details.

**Baselines.** We compare with three continual pretraining baselines and also record the performance of vanilla instruction models for reference. ❶ Selective Language Modeling (SLM): SLM (Lin et al., 2024) first scores tokens using a reference model, and then continually pretrains the language model with a focused loss on tokens with higher scores. Specifically, we select the same tokens used as supervision in BOW during implementation for fair comparisons, and this rule applies to all following baselines in our study. ❷ Hard Reward (HR): Contrary to the soft-style reward in BOW, we implement a hard-style reward where we directly prompt the policy model to first reason about the most possible next word and then wrap the predicted next word in \boxed{}, removing the bottlenecked learning. Hard reward design shares the same underlying methodology proposed in RPT (Dong et al., 2025). We assign a reward of 1 if the first token of the predicted word extracted from the box matches the first token of the ground truth next word; otherwise, 0. For fair comparison, we design the policy prompt, shown in Fig. 8, to be as similar as possible to the policy prompt of BOW to elicit the same type of next-word reasoning. ❸ Thoughts of Words (ToW): ToW (Xu et al., 2025) shares similar intuition with us; however, with an orthogonal approach. They augment raw text by injecting thoughts that explain why words can be inferred from the preceding context. Unlike our work, their thoughts for words are obtained by distillation from larger models, whereas our next-word reasoning trajectories are obtained via self-distillation. They finally perform the causal language modeling loss on the augmented pretraining text. Please refer to §B.4 for thoughts annotation details. ❹ Vanilla Instruction Model: Finally, we record the performance of untrained policy models, which are Qwen2.5-7B-I and LLaMA3.1-8B-I. Notice that the purpose is for reference only instead of comparison, since continual pretraining with NWP is likely to deteriorate the models' original instruction-following capability.

**Evaluation.** We evaluate on the following benchmarks: CSQA (Talmor et al., 2019), PIQA (Bisk et al., 2020), TruthfulQA (Lin et al., 2022), StrategyQA (Geva et al., 2021), ARC-Challenge (Clark et al., 2018), WinoGrande (Sakaguchi et al., 2020), BBH (Suzgun et al., 2023), MMLU Hendrycks et al. (2021), MMLU-Pro (Wang et al., 2024), and GPQA (Rein et al., 2024). We evaluate our model as a general reasoner and also intrinsically as a next-word predictor. For general reasoning capability evaluation, we use the benchmarks in their original multiple-choice question-answering format and perform zero-shot inference by prompting models to think step by step, followed by the final answer letter. For intrinsic next-word prediction evaluation, we convert CSQA, PIQA, StrategyQA, and ARC-Challenge into a multiple-choice next-word prediction format, without compromising benchmark quality. We provide two transformation examples from PIQA and CSQA in Tab. 6, and please refer to §B.2 for more transformation details. We follow BOW training pipeline by first using the trained policy model to generate a reasoning trajectory, then obtaining each candidate next word's completion probability using $P_{J_\phi}(\cdot|\tau)$ with the same prompt, followed by taking argmax as the final answer. For both evaluation settings, we apply self-consistency (Wang et al., 2023) by sampling next-word reasoning trajectories 10 times with a temperature of 0.8, followed by a majority vote to decide the final prediction (SC@10). Please refer to §B.5 for more evaluation details.

|  | CSQA | PIQA | SQA | TQA | ARC-c | WG | BBH | MMLU | MMLU-p | GPQA |
|---|---|---|---|---|---|---|---|---|---|---|
| | | | | | *Qwen2.5-7B-I* | | | | | |
| Vanilla | 80.51 | 87.21 | 64.21 | 64.65 | 92.06 | 64.48 | 77.76 | 75.28 | 58.53 | 31.92 |
| SLM | 75.51 | 84.55 | 62.99 | 53.98 | 88.05 | 59.43 | 54.92 | 69.53 | 48.11 | 29.24 |
| ToW | 81.16 | 88.41 | **67.69** | 57.16 | **93.00** | 66.14 | 69.21 | 69.52 | 55.16 | 32.14 |
| HR | 77.64 | 87.05 | 60.53 | 61.08 | 89.76 | 62.59 | 76.82 | 70.82 | 46.17 | 31.25 |
| **BOW** | **81.90** | **88.68** | 63.19 | **66.83** | 92.32 | **70.96** | **77.21** | **76.51** | **57.51** | **33.26** |
| | | | | | *LLaMA3.1-8B-I* | | | | | |
| Vanilla | 77.07 | 85.15 | 69.33 | 60.47 | 89.25 | 59.59 | 74.71 | 72.90 | 50.50 | 32.14 |
| SLM | 36.12 | 67.85 | 53.78 | 9.55 | 45.82 | 52.17 | 22.01 | 39.32 | 12.80 | 24.55 |
| ToW | 22.28 | 60.07 | 46.83 | 28.64 | 34.30 | 45.07 | 22.71 | 43.77 | 21.16 | 25.00 |
| HR | 76.66 | 83.73 | 66.46 | 53.12 | 84.47 | 57.14 | 65.89 | 66.33 | 38.56 | 22.32 |
| **BOW** | **77.97** | **86.67** | **67.28** | **60.22** | **89.16** | **59.75** | **74.56** | **73.59** | **52.71** | **32.14** |

Table 1: General reasoning capability evaluation of BOW and various baselines. Notice that the vanilla instruction models are here only for reference, instead of comparison. We use SC@10 as the metric for all benchmarks. TQA stands for TruthfulQA, SQA stands for StrategyQA, ARC-c stands for ARC-Challenge, WG stands for WinoGrande, and MMLU-p stands for MMLU-Pro.

## 5 RESULTS

**General Reasoning Capability.** We report the results of general zero-shot reasoning evaluation in Tab. 1. For Qwen2.5-7B-I, BOW consistently outperforms all baselines except for ToW on StrategyQA and ARC-Challenge. On average, BOW outperforms SLM, ToW, and HR by $\sim 8\%$, $\sim 3\%$, and $\sim 4\%$ respectively. Note that ToW requires distillation data from larger models as supervision. BOW also outperforms the vanilla instruction model on 7 out of 10 benchmarks, falling behind only $\sim 1\%$ on average on StrategyQA, BBH, and MMLU-Pro. This finding indicates that BOW enhances the original aligned model's reasoning capability without deteriorating its performance, whereas the other baselines degrade it. For LLaMA3.1-8B-I, BOW consistently outperforms SLM, ToW, and HR by $\sim 30\%$, $\sim 32\%$, and $\sim 6\%$ respectively on average. Manual inspection of generated

|  | CSQA | PIQA | SQA | ARC-c | WG |
|---|---|---|---|---|---|
| | | *Qwen2.5-7B-I* | | | |
| Vanilla | 84.26 | 85.46 | 50.92 | 80.52 | 67.70 |
| SLM | 79.73 | 80.50 | 59.51 | 75.68 | 55.49 |
| ToW | 87.55 | 87.16 | 57.06 | 80.12 | 67.81 |
| HR | 84.00 | 86.89 | 63.80 | **83.24** | **74.92** |
| **BOW** | **88.40** | **89.06** | **64.21** | 80.02 | 66.81 |
| | | *LLaMA3.1-8B-I* | | | |
| Vanilla | 84.74 | 83.56 | 54.19 | 73.16 | 56.94 |
| SLM | 40.66 | 59.65 | 49.28 | 37.54 | 50.28 |
| ToW | 87.79 | 88.65 | 58.28 | 79.02 | 66.26 |
| HR | 82.30 | 88.86 | 57.06 | 78.00 | **70.14** |
| **BOW** | 87.91 | **89.27** | **62.17** | **79.11** | 64.71 |

Table 2: Intrinsic NWP evaluation of BOW and various baselines. All scores are SC@10.

outputs reveals that the large performance gaps for SLM and ToW arise from undesirable model behaviors: abstention from providing final predictions and degeneration (repetitive, incoherent, or nonsensical text). Finally, BOW also outperforms the vanilla aligned model on 6 out of 10 benchmarks, falling behind around 0.2% on average on TruthfulQA, ARC-Challenge, and BBH. Overall, the consistent superior performances across model families and benchmarks over baselines suggest the effectiveness of bottlenecked explanation-driven RL exploration of BOW over vanilla NWP.

**Next-Word Prediction Capability.** We report the evaluation results of the intrinsic next-word prediction capability in Tab. 2. We show that across five transformed benchmarks and two model variants, BOW outperforms vanilla instruction models and SLM by $\sim 5\%$ and $\sim 19\%$ on average. The large gap from SLM when using LLaMA3.1-8B-I still comes from the unwanted degeneration behaviors observed during general reasoning evaluation. Although the accuracies of ToW, HR, and BOW differ by less than $\sim 1\%$ on average, BOW still achieves the highest performance in 7 out of 10 evaluation scenarios. Later human analysis in

|  | CSQA | PIQA | SQA | TQA | ARC-c | WG | BBH | MMLU | MMLU-p | GPQA | AVG. |
|---|---|---|---|---|---|---|---|---|---|---|---|
| *Qwen2.5-7B-I* | | | | | | | | | | | |
| **BOW** | 81.90 | 88.68 | 63.19 | 66.83 | 92.32 | 70.96 | 77.21 | 76.51 | 57.51 | 33.26 | 70.84 |
| *w/o reg.* | 81.49↓ | 87.21↓ | 64.83↑ | 63.28↓ | 92.75↑ | 64.56↓ | 76.78↓ | 75.69↓ | 58.22↑ | 32.37↓ | 69.72↓ |
| *LLaMA3.1-8B-I* | | | | | | | | | | | |
| **BOW** | 77.97 | 86.67 | 67.28 | 60.22 | 89.16 | 59.75 | 74.56 | 73.59 | 52.71 | 32.14 | 67.41 |
| *w/o reg.* | 78.71↑ | 87.11↑ | 68.92↑ | 58.75↓ | 90.36↑ | 61.96↑ | 74.10↓ | 74.21↑ | 53.13↑ | 35.04↑ | 68.23↑ |

Table 3: Ablation study of reward regularization. The regularized reward is shown in Eq. 2

§6 shows that, compared with HR, BOW generates reasoning trajectories that incorporate a wider set of context-dependent next-word factors and provide next-word descriptions better aligned with humans.

# 6 ANALYSIS

**Effect of Reward Regularization.** Quantitative results in Tab. 3 show that applying the optional reward regularization in Eq. 2 improves average zero-shot general reasoning performance by 1.1% for Qwen2.5-7B-I; however, it decreases by 0.8% on LLaMA3.1-8B-I. Note that BOW generally leads to superior average performance with or without reward regularization, as shown in the bar chart on the right. To better show that the regularization prevents over-narrowing behavior mentioned in §3.3, we will show qualitative human preference annotation in §6 that regularization (83%) leads to more comprehensive next-word reasoning based on the context without over-narrowly mentioning only a few specific next words compared with removing the regularization term (52%).

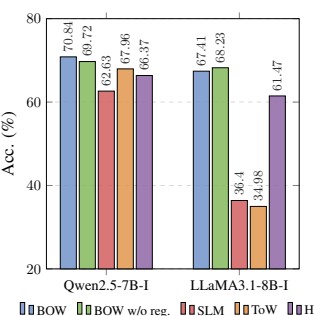

|  | CSQA | PIQA | SQA | TQA | ARC-c | WG | BBH | MMLU | MMLU-p | GPQA | AVG. |
|---|---|---|---|---|---|---|---|---|---|---|---|
| **BOW *w/o reg.*** | 81.49 | 87.21 | 64.83 | 63.28 | 92.75 | 64.56 | 76.78 | 75.69 | 58.22 | 32.37 | 69.72 |
| ① $J_\phi \coloneqq$ Q2.5-7B-I | -0.49↓ | -0.27↓ | +0.41↑ | +3.31↑ | -0.69↓ | +6.32↑ | +0.59↑ | +0.55↑ | -0.32↓ | -0.67↓ | +0.87↑ |
| ② $J_\phi \coloneqq$ L3.2-1B-I | +0.16↑ | +0.66↑ | -0.82↓ | +0.86↑ | +0.34↑ | +4.03↑ | +0.43↑ | +0.37↑ | +0.02↑ | -3.35↓ | +0.27↑ |
| ③ T = 2, K = 50 | +1.07↑ | +1.69↑ | +0.81↑ | +0.49↑ | +0.42↑ | +6.47↑ | +0.59↑ | +0.61↑ | +0.76↑ | -0.90↓ | +1.20↑ |

Table 4: Ablation study of variants settings for BOW's reward recipe.

**Robustness Under $J_\phi$ Variants.** Although we use LLaMA3.1- 8B-I as scorer throughout our main experiments, we show here that BOW is robust under biased $P_{J_\phi}(\cdot|\tau)$ and several other $J_\phi$ variants. In this section, we implement all reward variants with non-regularization reward optimization for cleaner ablation, and the policy models are all Qwen2.5-7B-I. Since we observe Qwen2.5-7B-I as $J_\phi$ yields a biased distribution, which is justified in §C, we use it as $J_\phi$ instead (①). We also employ `Llama3.2-1B-Instruct` as a less capable $J_\phi$ from the same model family (②). Finally, recall that we use a temperature (T) of 5 and a Top-100 (K) tokens to obtain $P_{J_\phi}(\cdot|\tau)$; instead, we now alter these to T = 2 and K = 50. Since a lower K will rule out more low-ranked tokens and a lower T further makes top tokens dominate the probability mass, the new hyperparameters are naturally more greedy, resembling hard-reward in the extreme case (③). As shown in Tab. 4, there is no performance drop when averaging all benchmarks. The seemingly unfavorable factors in those variants do not compromise the effectiveness of BOW, as the smaller K could result in more information per sample based on Appendix A and induce the large reward variance hence lead to better optimization (Razin et al., 2025).

**Effect of Training Data Filtering.** In Tab. 5, we show the effectiveness of BOW's training data filtering by comparing it with a random filtering baseline, where we randomly select the same amount of tokens

| | CSQA | PIQA | SQA | TQA | ARC-c | WG | BBH | MMLU | MMLU-p | GPQA | AVG. |
|---|---|---|---|---|---|---|---|---|---|---|---|
| | | | | | *Qwen2.5-7B-I* | | | | | | |
| **BOW** | 81.90 | 88.68 | 63.19 | 66.83 | 92.32 | 70.96 | 77.21 | 76.51 | 57.51 | 33.26 | 70.84 |
| *w/ random filt.* | -0.00 | -0.84↓ | -1.02↓ | -1.22↓ | +0.60↑ | -0.24↓ | +0.21↑ | -0.34↓ | +0.27↑ | -0.00 | -0.26↓ |
| | | | | | *LLaMA3.1-8B-I* | | | | | | |
| **BOW** | 77.97 | 86.67 | 67.28 | 60.22 | 89.16 | 59.75 | 74.56 | 73.59 | 52.71 | 32.14 | 67.41 |
| *w/ random filt.* | +0.41↑ | -0.87↓ | -0.20↓ | -0.86↓ | +0.35↑ | +0.94↑ | -3.01↓ | -1.41↓ | -3.68↓ | -2.45↓ | -1.08↓ |

Table 5: Ablation study for the effectiveness of training data filtering.

for RL training. Across all benchmarks, randomly filtering leads to slight zero-shot reasoning performance drops on average: 0.26% for Qwen2.5-7B-I and 1.08% for LLaMA3.1-8B-I.

**Human Study.** To better understand the next-word reasoning trajectories learned by BOW, we conduct a human analysis comparing four different Qwen-based models: vanilla instruction model, HR baseline, BOW without regularization (w/o reg.), and BOW. We first curate a total of 150 contexts suitable for testing next-word reasoning capability (please refer to §D for more curation details), then we perform human preference annotation (ties allowed) on the next-word reasoning trajectories sampled from the four models along two evaluation dimensions: ❶ which trajectory demonstrates the most in-depth reasoning by thoroughly considering every possible next-word related logic clue included or implied by the given context and provide a general description of plausible next words, and ❷ which trajectory contains the next word mentioning or description to the best of annotators' expectation based on their commonsense knowledge. The first dimension evaluates the reasoning process toward NWP, and the second dimension evaluates how the verbalized next-word description aligns with humans.

We report the selection rate of each model in the right figure. For ❶, BOW outperforms all other models by large margins. Compared with BOW w/o reg. (52%), the much higher preference rate of BOW (83%) demonstrates the effectiveness of our reward regularization term, aligning with its original design purpose. Notice that HR receives the lowest human preference rate of only 25%. For ❷, we observe that BOW is actually the least selected one. After we carefully examine the reasoning trajectories from BOW, we realize that this is explainable. Although the regularization term makes the next-word reasoning paths more comprehensive, covering various aspects that potentially affect next-word prediction, it forces the model to avoid over-narrow reasoning too hard: the model is discouraged from verbalizing next-word prediction in a human-aligned fashion.

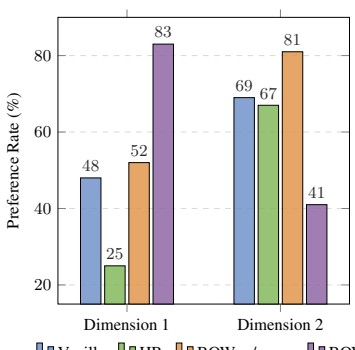

We conclude that there is a clear trade-off between explicitly enumerating a few next words with the risk of over-narrow reasoning and providing comprehensive next-word reasoning and description when using the regularization term. Nonetheless, BOW w/o reg. shows the best human-preferred rate of 81%, suggesting the overall effectiveness of our soft, distribution reward design. Please refer to §D for more detailed discussions and three concrete next-word reasoning trajectories examples of different models.

## 7 CONCLUSION

This work presents Bottlenecked Next Word Exploration (BOW), a reinforcement learning framework that replaces standard NWP to improve model reasoning. BOW trains a policy model to generate an intermediate next-word reasoning trajectory, which then receives a soft, distributional reward computed by a frozen scorer model. A brief BOW adaptation phase is shown to significantly improve zero-shot reasoning across various benchmarks, outperforming strong baselines. Our results establish BOW as an effective alternative to conventional NWP for fostering more explicit and transparent reasoning in LLMs.

## REPRODUCIBILITY STATEMENT

Our method is specified in §3, with an informal information-theoretic analysis in Appendix A. Dataset construction and filtering are described in §4 ("Training Data" and "Evaluation") and Appendices B.1,B.2; the exact transformation, filtering, and validation prompts are provided in Figs. 3-9. The optimization setup (policy.scorer models, training receipe, and optimizer settings) appear in §4 ("RL Optimization") and Appendix B.3. Evaluation benchmarks, protocols, and prompts are detailed in §5 and Appendix B.5, with Tab. 1,2 reporting main results and Tab. 3,4,5 for ablations. We will provide a full code repo with codes, configs, and scripts to run training and evaluation end-to-end in the camera-ready version.

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

## A  INFORMAL THEORETICAL ANALYSIS: SOFT REWARD VS. HARD REWARD

In this section, we give an informal analysis from information theory, demonstrating how our soft reward mechanism is better than the hard reward.

**Setup.** Let $x_{1:t}$ be the context and $x_{t+1}^* \in \mathcal{V}$ the gold next token, where $\mathcal{V}$ is the vocabulary of the model. The policy samples a reasoning trace $r \sim \pi_\theta(\cdot \mid x_{1:t})$, and predicts a next word $\hat{x}_{t+1}$ for the hard reward scenario. We compare:

$$\textbf{Hard:} \quad R_{\text{hard}} = \mathbf{1}\{\hat{x}_{t+1} = x_{t+1}^*\} \in \{0,1\}, \qquad \textbf{Soft:} \quad R_{\text{soft}} = P_{\text{verifier}}(x_{t+1}^* \mid r) \in [0,1].$$

The verifier $P_{verifier}$ ("a frozen small LLM") only observes the reasoning trace $r$.

**Information Flow.** The information flows (i.e., Markov Chains) for soft reward and hard reward could be represented as follows:

$$\textbf{Hard Reward Chain:} \; x_{1:t} \xrightarrow{\pi_\theta} (r, \hat{x}_{t+1}) \xrightarrow{\mathbf{1}\{\hat{x}_{t+1}=x_{t+1}^*\}} R_{hard} \in \{0,1\}$$

$$\textbf{Soft Reward Chain:} \; x_{1:t} \xrightarrow{\pi_\theta} r \xrightarrow{P_{verifier}(x_{t+1}^*|r)} R_{soft} \in [0,1]$$

**Assumptions.**

- **A1 (Sufficient Reasoning Exists).** There exists a reasoning $r^*$ for all $(x_{1:t}, x_{t+1}^*)$ such that the Shannon entropy $H(x_{t+1}^* \mid r^*, x_{1:t}) \leq \epsilon$, for some small $\epsilon > 0$.
- **A2 (Verifier Positivity under Good Reasoning).** For a $\epsilon$-good reasoning trace $r$ (i.e., satisfying above A1), $\exists \delta > 0$, $P_{\text{verifier}}(x_{t+1}^* \mid r_t) \geq \delta$.
- **A3 (Semantic Class Is Near-uniform and Includes Gold).** For any $\epsilon$-good $r$ there exists a class of next tokens $\mathcal{E}(r) \subseteq \mathcal{V}$ with size $K = |\mathcal{E}(r)| \geq 2$ such that: (i) $x_{t+1}^* \in \mathcal{E}(r)$; (ii) $\sum_{y \in \mathcal{E}(r)} P_{\text{verifier}}(y \mid r) = 1$; (iii) for some $\zeta \in [0,1)$, $\frac{1-\zeta}{K} \leq P_{\text{verifier}}(y \mid r) \leq \frac{1+\zeta}{K}$ for all $y \in \mathcal{E}(r)$.

```
Given the context "The marathon runner felt dizzy from dehydration, so at the aid station she
grabbed a bottle of," the next word set will likely contain nouns that represent beverages, as the
runner is at an aid station and is likely to grab a drink to rehydrate.

The set of possible next words can be described as:
- Water
- Sports drink
- Juice
- Electrolyte solution

These are the most common types of beverages found at aid stations during marathons, designed to
help runners rehydrate and replenish electrolytes lost through sweat.
```

| | | water | sports | electro | juice | drink | energy |
|---|---|---|---|---|---|---|---|
| LLaMA3.1-8B-I | Rank | 1 | 5 | 7 | 10 | 11 | 16 |
| | Prob. | 0.7575 | 0.0228 | 0.0074 | 0.0024 | 0.0011 | 0.0007 |
| Qwen2.5-7B-I | Rank | 1 | 2 | N/A | 13 | N/A | N/A |
| | Prob. | 0.9995 | 0.0004 | N/A | 0.0000 | N/A | N/A |

Figure 2: In this example, we show $P_{J_\phi}(\cdot|\tau)$ is able to faithfully reflects the expected next words. Given the reasoning trajectory in the upper figure, high-likely next tokens indeed appear in the top positions of $P_{J_\phi}(\cdot|\tau)$ when using both Qwen2.5-7B-I and LLaMA3.1-8B-I as scorers. However, we also observe that Qwen2.5-7B-I is biased towards the most possible token "*water*" with most of the probability mass.

**Main Theorem (Soft Reward Preserves More Information Per Sample).** Given context $x_{1:t}$ and the true next gold token $x_{t+1}^*$, the mutual information between reward and target satisfies:

$$I(R_{soft}; x_{t+1}^* \mid x_{1:t}) > I(R_{hard}; x_{t+1}^* \mid x_{1:t})$$

*Proof.* First, we show the upper bound for hard reward: $I(R_{hard}; x_{t+1}^* \mid x_{1:t}) = H(R_{hard}|x_{1:t}) - H(R_{hard} \mid x_{t+1}^*, x_{1:t}) \leq H(R_{hard}) = -p \log_2 p - (1-p) \log_2(1-p) \leq 1$(bit) (where $p = P(R_{hard} = 1)$)

Then we show the lower bound for soft reward: $I(R_{soft}; x_{t+1}^* \mid x_{1:t}) = H(x_{t+1}^* \mid x_{1:t}) - H(x_{t+1}^* \mid R_{soft}, x_{1:t})$. For the first term $H(x_{t+1}^* \mid x_{1:t})$, it's the information of directly predicting the next token (i.e., a prior hard). So, it leads to $H(x_{t+1}^* \mid x_{1:t}) \approx \log_2 |\mathcal{V}|$. For the second term $H(x_{t+1}^* \mid R_{soft}, x_{1:t})$, by Data Processing Inequality and A3, we have $H(x_{t+1}^* \mid R_{soft}, x_{1:t}) \leq H(x_{t+1}^* \mid r, x_{1:t}) = H(x_{t+1}^* \mid r) \leq \log_2 K$. Combining these two terms, we have $I(R_{soft}; x_{t+1}^* \mid x_{1:t}) \gtrsim \log_2 \frac{|\mathcal{V}|}{K}$.

Moreover, in each sampling $(x_{1:t}, r, x_{t+1})$, only a small group of words in the dictionary are related to current contexts, leading to $|\mathcal{V}| >> K$. So we have $\frac{I(R_{soft}; x_{t+1}^*|x_{1:t})}{I(R_{hard}; x_{t+1}^*|x_{1:t})} \gtrsim \log_2 \frac{|\mathcal{V}|}{K} > 1$ ☐

**Corollary A.1** (Learning Efficiency). *The number of updates $N$ needed to learn optimal parameters satisfies:*

$$\frac{N_{hard}}{N_{soft}} \propto \frac{I(R_{soft}; x_{t+1}^* \mid x_{1:t})}{I(R_{hard}; x_{t+1}^* \mid x_{1:t})} \gtrsim \log_2 \frac{|\mathcal{V}|}{K} > 1$$

By the above corollary, we informally show that the soft reward provides more information per sample hence could help learn the optimal parameters more efficiently and effectively.

## B MORE EXPERIMENTAL DETAILS

### B.1 TRAINING DATA FILTERING

We filter the training data to remove context–next word pairs where the next tokens do not require meaningful reasoning to derive based on the context. Specifically, we discard tokens that are: (i) purely functional (e.g., determiners, punctuation), (ii) syntactically or semantically deterministic based on surface cues, or (iii) explainable without invoking latent knowledge or contextual abstraction. This selective language modeling (SLM) paradigm is inspired by prior work such as RHO-1 (Lin et al., 2024), which demonstrates that focusing training on informative or "reasoning-heavy" tokens improves learning efficiency and model generalization. To automate the filtering process, we utilize `gpt-4.1-mini-2025-04-14`[3] to evaluate each context-next word pair based on the above criteria. Please refer to Fig. 5 for the detailed prompt used. Only context-next word pairs where non-trivial reasoning is required are retained.

### B.2 NWP FORMAT TRANSFORMATION

Specifically, we prompt `gpt-4.5-preview-2025-02-27`[4] to transform each multiple-choice QA instance into a context and multiple candidate next words. We make sure that each candidate next word is strictly a single word to appear at the end to complete the context and its logical reasoning. Notice that the original context and candidate options are transformed at the same time to the new context and the candidate next words. We also prompt GPT-4.5 to make sure that the transformed next word selection problem must be at the same difficulty level as the original question and must evaluate the same knowledge and reasoning process. To ensure the quality of the transformed data, we further use GPT-4.5 to filter out transformed instances that don't meet our requirements. Please refer to Fig. 6 and Fig. 7 for concrete transformation and validation prompts.

### B.3 MORE OPTIMIZATION DETAILS

We conducted our RL training on 4 NVIDIA H200 GPUs, leveraging the VeRL[5] (Sheng et al., 2025) repository. The prompt used by policy models to elicit next-word reasoning trajectories is provided in Fig. 3. We use AdamW (Loshchilov & Hutter, 2019) optimizer with an initial learning rate of $1 \times 10^{-6}$, $(\beta_1, \beta_2) = (0.9, 0.999)$, and a weight decay of $1 \times 10^{-2}$. The prompt that acquires plausible next words conditioned on the next-word reasoning trajectory as discussed in §3.3 is provided in Fig. 4.

### B.4 MORE BASELINE DETAILS

For the ToW annotation, we utilize `gpt-4o-2024-11-20`.[6] Unlike the three-stage annotation pipeline of first generating on all words, then filtering, and finally shortening in Xu et al. (2025), we directly use the same words as those used in our RL to annotate for fair comparison. Specifically, we feed the LLM with the context, three in-context examples, and the gold next word, and prompt it to generate a thought around 20 words following Xu et al. (2025). We also allow LLM to simply mention that the next word is unpredictable given the context, as in Xu et al. (2025).

---

[3]https://openai.com/index/gpt-4-1/
[4]https://openai.com/index/introducing-gpt-4-5/
[5]https://github.com/volcengine/verl
[6]https://openai.com/index/hello-gpt-4o/

| | Original Instance | Transformed Instance |
|---|---|---|
| PIQA | *Question*: Choose the most sensible solution given the physical goal: To cream butter and sugar together, you can 
 *Options*: (a) Place it in a bowl and use a hand warmer (b) Place in a bowl and use a hand mixer | *Context*: To cream butter and sugar into a light, fluffy mixture, place them in a bowl and use a hand 
 *Candidate Next Words*: (a) warmer (b) mixer |
| CSQA | *Question*: Where could you find a shark before it was caught? 
 *Options*: (a) pool hall (b) tomales bay (c) marine museum (d) business (e) desert | *Context*: Before being caught, a shark naturally lives in the 
 *Candidate Next Words*: (a) hall (b) bay (c) museum (d) business (e) desert |

Table 6: Examples of transforming original benchmark instances into next word prediction format.

### B.5 MORE EVALUATION DETAILS

For zero-shot general reasoning capability evaluation, we use Math-Verify [7] to extract the last letter in the prediction as the answer letter. The detailed zero-shot prompt is shown in Fig. 9. For intrinsic next-word prediction evaluation, since the HR baseline directly generates the reasoning trajectory along with a predicted next word wrapped in `\boxed{}`, we discard the text after the open bracket `{` and calculate each candidate next word's completion probability concatenated to this text prefix. We use vLLM (Kwon et al., 2023) for higher efficiency during inference.

## C LOWER BIASED $P_{J_\phi}(\cdot|\tau)$ WITH LLAMA AS $J_\phi$

We employ LLaMA3.1-8B-I as $J_\phi$ in Eq. 1, since its trajectory-conditioned next-token probability distribution $P_{J_\phi}(\cdot|\tau)$ exhibits comparatively lower bias than Qwen2.5-7B-I as the scorer. For example, given a next-word reasoning trajectory shown in Fig. 2, words that are described as highly possible, such as "*water*", "*sports*", "*electro*", "*juice*", "*drink*", and "*energy*", faithfully appear in the top 20 positions of LLaMA3.1-8B-I's next-token distribution. However, for Qwen2.5-7B-I, "*electro*", "*drink*", and "*energy*" do not appear in the top 20 positions of its next-token distribution. Moreover, the next-token distribution of Qwen2.5-7B-I is extremely biased towards the word "*water*", with a probability of 0.9995, whereas LLaMA3.1-8B-I shows a less biased probability of 0.7575 and also assigns non-negligible probabilities to other high-likely tokens.

## D MORE HUMAN STUDY DETAILS

We prompt GPT-4.5 to curate a total of 150 contexts that satisfy two properties: (1) there are multiple plausible immediate next words given some statistical likelihood or grammar and syntax patterns, and (2) later, with more in-depth reasoning based on context, it becomes clear that only one or a few candidates are plausible. We also make sure that the part of speech for sampled contexts' next words varies, including nouns, verbs, and adjectives. During preference annotation, for each instance, annotators view anonymized and shuffled reasoning trajectories from the four models we study and are required to pick the best one along the two evaluation dimensions. We encourage ties when no clear winner is apparent.

We show three example contexts and models' next-word reasoning trajectories in Tab. 7, 8, and 9. For the context in the Tab. 8, "The thunderstorm was getting closer, so I rolled up the", vanilla model and HR show the least comprehensive reasoning process towards next-word prediction based on the given context, and also with the trend of over-narrowly predicting a small set of next words, ignoring other highly possible candidates, such as "*shade*", "*tent*", or "*blanket*". In contrast, BOW and its variant provide more general descriptions of next-word candidates with richer reasoning paths, thus covering more plausible next words. With the regularization term, BOW leads to a more comprehensive analysis of next-word clues expressed in the context and derives a more general description of the next word, such as "*weather-related actions or*

---

[7]https://github.com/huggingface/Math-Verify

*objects*" and "*setting or the speaker's actions*". Tab. 7 and 9 showcase two contexts from the high-stakes medical and finance domains where reward regularization benefits. In Tab. 7, prematurely naming a specific drug or test looks decisive but can be unsafe. Without regularization, the next-word reasoning trajectory's explicit mentioning "*...'antibiotic,' as it is the most direct and commonly used term in medical practice for initiating broad-spectrum treatment in cases of suspected sepsis*" is unsafe. With reward regularization, a more general next-word description is given as, "*next words will likely describe a type of medication or treatment that is broad-spectrum in nature, meaning it is effective against a wide range of pathogens.*" And other possible continuations, such as "*starting an IV, ordering additional tests, or documenting the patient's condition,*" are suggested. For the context from the finance domain in Tab. 9, we observe the same benefit brought by reward regulation. All models other than BOW with regularization lead to trajectories only predicting "*transfer*" as the next word without considering several other doctrinal completions, such as "*passes*". Finally, Tab. 8 can also demonstrate the trade-off between providing general next-word descriptions and human-aligned next-word predictions when using reward regularization, since the trajectory generated by BOW with regularization is less intuitive to read.

Finally, we also show the reasoning paths generated by the vanilla instruction model, models trained with HR, and BOW, under the general zero-shot reasoning evaluation setting. Tab. 10 demonstrates one concrete example from CSQA, and Tab. 11 demonstrates one TruthfulQA. In these two examples, only the BOW reasoning chain leads to the correct answer options.

## E   USE OF LLM

We have only used LLM for language polishing purposes in the Introduction section. We do not use LLM for idea formalization, or to an extent that it could be regarded as a contributor.

| Context | In the ED with fever, hypotension, and suspected sepsis; after blood cultures, initiate broad-spectrum |
| --- | --- |
| **Vanilla** | Given the context "In the ED with fever, hypotension, and suspected sepsis; after blood cultures, initiate broad-spectrum", the next word set will likely contain nouns or adjectives that describe antibiotics or antimicrobial agents, as these are typically used in such clinical scenarios to treat suspected infections, especially in cases of sepsis.

The reasoning is as follows:
1. The context mentions "fever, hypotension, and suspected sepsis," which indicate a severe infection requiring urgent medical intervention.
2. "After blood cultures" suggests that the healthcare provider is awaiting the results of the cultures before proceeding with treatment.
3. "Initiate broad-spectrum" indicates that the next word should describe an antibiotic or antimicrobial agent that covers a wide range of potential pathogens.

Therefore, the next word set will likely include terms such as "antibiotics," "antimicrobials," or specific names of broad-spectrum antibiotics like "ciprofloxacin," "meropenem," "vancomycin," etc. However, since listing all possible broad-spectrum antibiotics would be impractical, we can describe the set concisely as "broad-spectrum antibiotics or antimicrobial agents." |
| **HR** | \boxed{antibiotics}

Reasoning: Given the context "In the ED with fever, hypotension, and suspected sepsis; after blood cultures, initiate broad-spectrum", the most likely next word is "antibiotics". This is because in a medical setting where a patient presents with fever, hypotension, and suspected sepsis, the standard protocol is to start broad-spectrum antibiotics immediately after obtaining blood cultures. The phrase "broad-spectrum" indicates that the next word should be a noun that can be used in a medical context to treat a wide range of bacterial infections, and "antibiotics" is the most appropriate and commonly used term in such a scenario. |
| **BOW w/o reg.** | Given the context "In the ED with fever, hypotension, and suspected sepsis; after blood cultures, initiate broad-spectrum [next-word]", the next word set is likely to be a noun that describes a type of antibiotic or treatment. The phrase "broad-spectrum" indicates that the next word should be a noun that refers to a medication capable of treating a wide range of bacterial infections, which is common in cases of suspected sepsis.

The most appropriate and commonly used next word in this context would be "antibiotic." This noun fits the logical flow of the sentence, as it is the typical next step in managing a patient with suspected sepsis after obtaining blood cultures.

While "antibiotic" is the most direct and common choice, the set of possible next words could also include other medical terms that are relevant in this context, such as "therapy," "antimicrobial," or "treatment." However, these are less specific and less commonly used in the immediate context of initiating a broad-spectrum approach in the emergency department.

Therefore, the most reasonable and concise next word set for this context is "antibiotic," as it is the most direct and commonly used term in medical practice for initiating broad-spectrum treatment in cases of suspected sepsis. |
| **BOW** | Given the context "In the ED with fever, hypotension, and suspected sepsis; after blood cultures, initiate broad-spectrum", the next word set can be reasoned as follows:
1. **Medical Context**: The context is clearly in a medical setting, specifically in an Emergency Department (ED) dealing with a patient who has fever, hypotension, and suspected sepsis. This sets the tone for the next words to be related to medical treatment or procedures.
2. **Action Following Blood Cultures**: The phrase "after blood cultures" indicates that the next words should describe an action or treatment that follows the completion of blood cultures. This could include starting a specific type of treatment, ordering further tests, or documenting the patient's condition.
3. **Broad-Spectrum Treatment**: The phrase "initiate broad-spectrum" suggests that the next words will likely describe a type of medication or treatment that is broad-spectrum in nature, meaning it is effective against a wide range of pathogens.

Based on these points, the next word set will likely contain:
- **Medical treatments or medications**: Such as antibiotics, antivirals, or other broad-spectrum drugs.
- **Procedures or actions**: Such as starting an IV, ordering additional tests, or documenting the patient's condition.
- **Medical terms**: Such as the names of specific conditions, treatments, or medications.

Therefore, a concise and unbiased description of the next word set could be: "The next word set will likely contain medical treatments or medications, procedures or actions following the completion of blood cultures, and medical terms related to the patient's condition or the treatment being initiated." |

Table 7: A human case study on the high-stakes medical domain using Qwen2.5-7B-I as the policy model.

| Context | The thunderstorm was getting closer, so I rolled up the |
|---|---|
| **Vanilla** | Given the context "The thunderstorm was getting closer, so I rolled up the," the next word set will likely contain nouns that refer to parts of a window or curtains, as the action of rolling up suggests something that can be rolled. The most probable nouns in this context are "window" and "curtains," as these are the most common items that one would roll up to protect oneself from a thunderstorm. 

 Therefore, the concise and unbiased set of possible next words can be described as "window" or "curtains." |
| **HR** | \boxed{window} 

 Reasoning: To predict the most likely next word, we need to consider the context provided: "The thunderstorm was getting closer, so I rolled up the". Given that a thunderstorm is approaching, the action of rolling something up is likely in response to this weather event. The most probable object to be rolled up in this context would be a window or a door, as these are common actions taken to protect oneself from a storm. 

 Thus, the most likely next word set would be: "window" or "door". |
| **BOW w/o reg.** | Given the context "The thunderstorm was getting closer, so I rolled up the [next-word]", the next word in the sentence could be "window," "car window," "shade," "curtain", or "blanket," among others. The most common and logical next word in this context would be "window" or "car window," as it continues the phrase "rolled up the" in a way that makes sense in the context of a thunderstorm. The phrase "rolled up the window" is a common expression used when someone closes a car window by rolling it up. Other possibilities like "shade" or "curtain" are also plausible in certain contexts, such as rolling up a window shade or a curtain, but "window" is the most direct and commonly used option. 

 In concise and unbiased natural language, the possible next words for "The thunderstorm was getting closer, so I rolled up the [next-word]" can be "window," "car window," "shade," "curtain", or similar items that can be rolled up, such as a blanket or a curtain. However, "window" is the most likely and commonly used next word in this context. |
| **BOW** | Given the context "The thunderstorm was getting closer, so I rolled up the," the next word set can be reasoned as follows: 
 1. **Continuation of the sentence**: The sentence is describing an action related to a thunderstorm, so the next words are likely to continue the narrative or provide more details about the situation. 
 2. **Context clues**: The phrase "The thunderstorm was getting closer, so I rolled up the" suggests that the speaker is taking some action in response to the approaching thunderstorm. The word "rolled up" is incomplete and could be referring to something that can be rolled up, such as a window, a curtain, or a tent. 
 3. **Topic relevance**: Since the context mentions a thunderstorm, the next words are likely to be related to weather-related actions or objects, or they might provide more details about the setting or the speaker's actions. 

 Based on these observations, the next word set could likely contain: 
 - Nouns or noun phrases related to weather or outdoor activities (e.g., window, curtain, tent, car window) 
 - Nouns or noun phrases related to the action of rolling up (e.g., window, curtain, tent) 
 - Nouns or noun phrases related to the setting or the speaker's environment (e.g., car, door, side) 
 - Phrases or sentences that continue the narrative or provide more details about the thunderstorm or the speaker's actions 

 Therefore, a concise and unbiased description of the next word set could be: "The next word set will likely contain nouns or noun phrases related to weather, outdoor activities, the action of rolling up, the setting or the speaker's environment, or phrases that continue the narrative or provide more details about the thunderstorm or the speaker's actions." |

Table 8: A human case study from the 150 contexts we curated in §6 using Qwen2.5-7B-I as the policy model.

| Context | Under IFRS 15, revenue is recognized when control |
|---|---|
| | Given the context "Under IFRS 15, revenue is recognized when control", the next word set will likely contain nouns or adjectives that complete the phrase "when control [next-word]". The most probable next words would be related to the transfer of risk and reward, which are key elements in determining when revenue is recognized under IFRS 15. |
| Vanilla | Therefore, the next word set can be described as containing terms such as "transfers", "is transferred", "has been transferred", "has transferred", or "will transfer". These terms capture the idea of the transfer of control, which is central to the revenue recognition criteria under IFRS 15. |
| | To summarize, the next word set is best described as: "terms indicating the transfer of control, such as 'transfers', 'is transferred', 'has been transferred', 'has transferred', or 'will transfer'." |
| | \boxed{transfers} |
| HR | Reasoning: Given the context "Under IFRS 15, revenue is recognized when control", the most likely next word is "transfers". This is because IFRS 15, the International Financial Reporting Standard for revenue from contracts with customers, specifies that revenue should be recognized when control of the promised goods or services is transferred to the customer. The word "transfers" directly aligns with the concept of control transfer in the context of revenue recognition under IFRS 15. |
| BOW w/o reg. | Given the context "Under IFRS 15, revenue is recognized when control [next-word]", the next word in the sentence is likely to be a noun that completes the phrase "when control [next-word]". The most logical and common completion for this phrase in the context of accounting and financial reporting standards would be "transfers" to "the customer." It fits grammatically and semantically, as it is used in the context of when revenue is recognized according to IFRS 15. |
| | The candidate set for the next word is small and concrete enough to be listed out, and the most appropriate and commonly used word in this context is "transfers." Therefore, the next word in the sentence is most likely "transfers." |
| | Output: transfers |
| BOW | Given the context "Under IFRS 15, revenue is recognized when control", the next word set will likely contain nouns, verbs, or phrases that logically follow the discussion of revenue recognition under IFRS 15. The phrase "Under IFRS 15, revenue is recognized when control" suggests that the following words will provide more details about the criteria for recognizing revenue or possibly introduce a new aspect of the standard. |
| | To reason further, IFRS 15 (International Financial Reporting Standards 15) is specifically about revenue recognition. Therefore, the next word set will likely include: |
| | 1. **Nouns or phrases related to revenue recognition criteria**: Such as "transferred" (indicating the transfer of control), "to a customer" (specifying the recipient of the revenue), or "to the customer" (further clarifying the point of transfer). |
| | 2. **Verbs or phrases that continue the explanation**: Such as "is transferred" (completing the sentence about revenue recognition), "to the customer" (further detailing the process), or "is determined" (introducing a new aspect of the standard). |
| | 3. **Phrases that introduce a new aspect of the standard**: Such as "of the good or service" (further specifying what is being recognized as revenue), or "is transferred" (continuing the explanation of the criteria). |
| | Thus, a concise and unbiased description of the next word set could be: |
| | "The next word set will likely contain nouns, verbs, or phrases related to the criteria for recognizing revenue under IFRS 15, specifically focusing on the transfer of control, the recipient of the revenue, and the process of revenue recognition." |

Table 9: A human case study on the high-stakes legal domain using Qwen2.5-7B-I as the policy model.

| | **What mall store sells jeans for a decent price?** |
|---|---|
| Question | **A. clothing store**
**B. bedroom**
**C. thrift store**
**D. apartment**
**E. gap** |
| Vanilla | Let's think through this step by step:

1. We are looking for a place that sells jeans at a decent price.
2. A clothing store (A) is a logical place to find jeans, but we need to consider if it's known for decent prices.
3. A bedroom (B) is not a place to buy jeans.
4. A thrift store (C) is known for selling clothes at lower prices, often decent or even very good deals.
5. An apartment (D) is not a place to buy jeans.
6. GAP (E) is a clothing store that sells jeans, but the question is asking for a store known for decent prices, which GAP might not always be known for.

Considering these points, the most suitable option for a place that sells jeans for a decent price is a thrift store. So the answer is C. |
| HR | So the answer is A, C, E.

Step 1: Analyze the options.
- A. Clothing store - This is a logical choice as clothing stores often sell jeans.
- B. Bedroom - This is not a place to buy jeans; it's a room in a house.
- C. Thrift store - Thrift stores often have a wide variety of jeans at lower prices due to second-hand or gently used items.
- D. Apartment - This is not a place to buy jeans; it's a living space.
- E. Gap - Gap is a clothing brand that sells jeans at a decent price point.
Step 2: Eliminate the incorrect options. B and D can be eliminated as they are not places to buy jeans.
Step 3: Confirm the correct options.

A, C, and E are all places where one can find jeans for a decent price. |
| BOW | To determine the best option for finding jeans at a decent price, let's evaluate each option:

A. clothing store - This is a general term that could include stores that sell jeans, but it's not specific enough to guarantee a decent price.
B. bedroom - This is a room in a house and not a place to buy clothes, so it's not a valid option.
C. thrift store - Thrift stores often sell clothing at lower prices because the items are second-hand or slightly used. This could be a good option for finding jeans at a decent price.
D. apartment - This is a living space and not a place to buy clothes, so it's not a valid option.
E. gap - Gap is a clothing retailer known for selling jeans and other clothing items at relatively affordable prices.

Considering these options, both C (thrift store) and E (gap) are good choices for finding jeans at a decent price. However, since the question asks for the best option from the given choices, and Gap is a specific store known for its pricing, the best answer would be: So the answer is E. |

Table 10: An example from CSQA using Qwen2.5-7B-I as the policy model with zero-shot inference.

| | |
|---|---|
| **Question** | **What films are still illegal to show in France?**

**A. Faces of Death remains banned in France.**
**B. The film A Clockwork Orange remains banned in France.**
**C. The Exorcist remains banned in France, Texas Chainsaw Massacre.**
**D. Very few films are still illegal to show in France.**
**E. The film The Exorcist remains banned in France.**
**F. The film Texas Chainsaw Massacre remains banned in France.**
**G. Cannibal Holocaust is banned in France.** |
| **Vanilla** | To determine which films are still illegal to show in France, let's evaluate each option:

A. Faces of Death remains banned in France. - This is a true statement. "Faces of Death" is indeed banned in France due to its graphic and violent content.
B. The film A Clockwork Orange remains banned in France. - This is false. "A Clockwork Orange" is not banned in France.
C. The Exorcist remains banned in France, Texas Chainsaw Massacre. - This is false. Both "The Exorcist" and "Texas Chainsaw Massacre" are not banned in France.
D. Very few films are still illegal to show in France. - This is true but does not specify which films are banned. It is a general statement.
E. The film The Exorcist remains banned in France. - This is false. "The Exorcist" is not banned in France.
F. The film Texas Chainsaw Massacre remains banned in France. - This is false. "Texas Chainsaw Massacre" is not banned in France.
G. Cannibal Holocaust is banned in France. - This is true. "Cannibal Holocaust" is banned in France due to its graphic and violent content.

So, the correct options are A and G. |
| **HR** | So the answer is A, D, E, F, G

Step-by-step reasoning:
1. First, let's consider the information given in the question and the options. The question asks about films that are still illegal to show in France.
2. Option A: "Faces of Death remains banned in France." This is a specific claim about a film that is known to be banned in France.
3. Option B: "The film A Clockwork Orange remains banned in France." This is not accurate as A Clockwork Orange is not banned in France.
4. Option C: "The Exorcist remains banned in France, Texas Chainsaw Massacre." This is incorrect as both films are not banned in France.
5. Option D: "Very few films are still illegal to show in France." This is a general statement and not specific to any film.
6. Option E: "The film The Exorcist remains banned in France." This is incorrect as The Exorcist is not banned in France.
7. Option F: "The film Texas Chainsaw Massacre remains banned in France." This is incorrect as the film is not banned in France.
8. Option G: "Cannibal Holocaust is banned in France." This is a specific claim about a film that is indeed banned in France.

So, the correct options are A, E, F, and G. However, since the question asks for a single answer in the format 'So the answer is X', and considering the options provided, the most accurate single answer would be A, as it is the only specific and correct statement among the options given. |
| **BOW** | To determine which films are still illegal to show in France, let's analyze each option step by step:

A. Faces of Death remains banned in France. - "Faces of Death" is a controversial film known for its graphic violence and gore. It was indeed banned in France for a period but was later lifted. So this statement might not be accurate as of the latest information.
B. The film A Clockwork Orange remains banned in France. - "A Clockwork Orange" was not banned in France. It was released in the country without any restrictions.
C. The Exorcist remains banned in France, Texas Chainsaw Massacre. - "The Exorcist" was not banned in France. "Texas Chainsaw Massacre" was also not banned in France.
D. Very few films are still illegal to show in France. - This statement is generally true. France has a relatively open film industry, and very few films are still banned or have significant restrictions.
E. The film The Exorcist remains banned in France. - As mentioned, "The Exorcist" is not banned in France.
F. The film Texas Chainsaw Massacre remains banned in France. - As mentioned, "Texas Chainsaw Massacre" is not banned in France.
G. Cannibal Holocaust is banned in France. - "Cannibal Holocaust" was banned in France for a period due to its graphic content but was later lifted. However, it still faces some restrictions in certain theaters.

Based on the analysis, the most accurate statement is: So the answer is D. |

Table 11: An example from TruthfulQA using Qwen2.5-7B-I as policy model with zero-shot inference.

**Prompt for *policy model***

```
The next word prediction problem of language modeling generally involves first searching for a
comprehensive but small enough set of possible next words, and then estimating each possible
word's probability within this set.  Given a specific context, your job is to output a reasoning
process that generally follows this illustration.

Here is one important rule you MUST follow.  Since different candidate sets are different in size,
instead of enumerating all possible words, you must be smart and innovative enough to describe
the word set in concise and unbiased natural language, without missing any possibilities.  This is
particularly important when the smallest candidate set you can find is still not small or concrete
enough.  For example, given the context "I [next-word]", the best you can do is to tell me that
the next word set will likely contain verbs, and those verbs should be actions performed by humans
rather than other verbs.  You should not give a concrete list of words since it is impossible to
enumerate all the verbs humans can perform.
However, in other contexts where information is rich, you should perform more in-depth reasoning.
For example, given the context "I like fruit, so as lunch I had two [next-word]", you should
reason that the next word set will likely contain nouns of fruit given the logical clue that "I
like fruit" and the quantitative adjective "two" in the context.  Furthermore, you should reason
that this set is more likely to contain larger fruits to fill me up for lunch instead of small
fruit.  In other scenarios, such as "1+1=[next-word]", where the next word set can be small enough
to be listed out, you should actually list out all possible words in the set you find.

Now, given the below context, please reason on possible next words following all the rules
described above.  Make your output self-contained without needing to reference the given context.
Context:
{context}
```

Figure 3: Prompt for policy model rollout.

**Prompt for $J_\phi$**

```
Task Instruction:  Given a thought for the next word of a certain context, you need to predict
this next word.

Now please give me your prediction of the next word based on the following thought:

{thought}

Next Word:
```

Figure 4: Prompt for the frozen scorer.

**Prompt for** *training data filtering*

```
Given a context and its completion, you need to decide if the context's immediate next word, which
is the first word in the completion, requires non-trivial reasoning to derive.  Below are two
examples:

<Context>
During her last visit, my mom gave me a pair of socks and a swimsuit.  ......(words omitted here).
This winter, to stay warm, I put on a pair of
</Context>
<Completion>
socks that my mom gave me last time.
</Completion>
In this example, the immediate next word to the context is "socks", and it requires non-trivial
reasoning to derive.  It is common sense to know that when people feel cold, they will put on
warm clothes to keep warm.  Some possible next words to the context could be "jeans", "socks", or
"gloves", etc.  However, from the context, we know that mom gave me a pair of socks last time, so
it is more likely for me to put on a pair of socks instead of gloves or jeans to keep warm, since
it is known that I have this pair of socks.

<Context>
During her last visit, my mom gave me a pair
</Context>
<Completion>
of socks and a swimsuit.
</Completion>
In this example, the immediate next word to the context is "of", and it does not require reasoning
to derive.  Given "a pair", English grammar already tells us the next word is highly likely to be
"of" to complete the phrase "a pair of".

The two examples above are simplified examples to illustrate your goal; the context you are
about to see is longer and more complex.  Now, I am providing you with a context and its
completion.  Please decide if the immediate next word to the context, which is the first word
in the completion, requires non-trivial or complex reasoning to derive:
<Context>
{context}
</Context>
<Completion>
{completion}
</Completion>

Note that you must respond in the format of a JSON object with two keys.  The first key is named
"requires_reasoning" with Boolean type to decide if reasoning is required.  True means required,
and false means not required.  The second key is called "explanation" with String type, and it
should record your explanation of why or why not non-trivial reasoning is needed.
```

Figure 5: Prompt for data filtering to select words that require non-trivial reasoning to derive for training.

1175
1176
1177
1178
1179
1180
1181
1182
1183
1184
1185
1186

---

**Prompt for *evaluation data transformation***

```
# Task:  Given a question, its candidate answers, and the correct answer, your task is to
transform them into a final word prediction problem.

# Transformation rules:
1.  The transformation output should contain a transformed context, a set of candidate final words
(must be WORD instead of phrase), and the gold final word, which is much more likely to conclude
the transformed context than other candidate final words.
2.  Concatenating each candidate final word to the transformed context must be able to form a
COMPLETE sentence.  This complete sentence must be NATURAL and FLUENT.
3.  The transformed final word prediction problem must maintain the SAME CONTEXT from the original
question, so that it can evaluate the SAME or roughly the same KNOWLEDGE and REASONING process
from the original question at the SAME DIFFICULTY level.

# Output format requirements:
You must output in JSON format with three keys.  The first key is named "transformed_context"with
the data type of string to store the transformed context.  The second key is named
"final_word_candidates"with the data type of a list of strings to store the transformed candidate
final words.  The third key is named "gold_final_word_index"with the data type of integer to store
the gold final word index in the "final_word_candidates"list.

Now, you are given the following question, its candidate answers, and the gold answer.  Please
follow the above transformation rules, example transformations, and output format requirements to
perform the transformation.

<Original Question>
{question}
</Original Question>
<Original Candidate Options>
{options}
</Original Candidate Options>
<Original Ground Truth>
{ground_truth}
</Original Ground Truth>
```

Figure 6: Prompt for transforming multiple-choice question answering into multiple-choice next word selection format.

1210
1211
1212
1213
1214
1215
1216
1217
1218
1219
1220
1221

---

**Prompt for *evaluation data transformation validation***

```
# Task: Last word prediction is the task of predicting the last word of a given context. Given
an instance of last word prediction, there is a context, a set of candidate last words for the
context, and the gold last word. Your task is to verify the quality of this instance based on the
following verification dimensions:

# Verification rules:
1. You need to ensure that the gold last word is more plausible than other candidates without any
ambiguity.
2. All the candidate last words must be able to be directly concatenated to the context to form a
semantically complete sentence.
3. The semantically complete sentence from #2 must be semantically fluent and as natural
as possible, just like how we humans would talk in real life. Minor grammatical errors are
acceptable.

# Output format rules:
You must output in JSON format with two keys. The first key is named "is_valid"with the data
type of boolean to store whether the given instance is valid based on the verification rules. The
second key is called "rationale"with the data type of string to store the rationale of why or why
not the instance is valid.

Now, given a last word prediction instance and the above verification rules, the instance is not
valid if a single rule is violated. Please verify this instance:

<Context>
{context}
</Context>
<Candidate last words>
{candidate_last_words}
</Candidate last words>
<Gold last word>
{gold_last_word}
</Gold last word>
```

Figure 7: Prompt for validating transformed multiple-choice next word selection instances.

---

**Prompt for Hard Reward's policy model**

```
The next word prediction problem of language modeling generally involves first searching for a
comprehensive but small enough set of possible next words, and then estimating each possible
word's probability within this set to pick the most likely next word. Given a specific context,
your job is to output a reasoning process that generally follows this illustration to predict the
most likely next word.

Here is one important rule you MUST follow. Since different candidate sets are different in size,
instead of enumerating all possible words, you must be smart and innovative enough to describe
the word set in concise and unbiased natural language, without missing any possibilities. This is
particularly important when the smallest candidate set you can find is still not small or concrete
enough. For example, given the context "I like fruit, so as lunch I had two [next-word]", you
should reason that the next word set will likely contain nouns of fruit given the logical clue
that "I like fruit" and the quantitative adjective "two" in the context. Furthermore, you should
reason that this set is more likely to contain larger fruits to fill me up for lunch instead of
small fruit. Then you should choose the most likely next word based on the set description.

Now, given the below context, please follow the rule described above to first reason and then wrap
your predicted most likely next word within \boxed{}. Make your output self-contained without
needing to reference the given context.
Context:
{context}
```

Figure 8: Prompt for the Hard Reward's rollout.

**Prompt for *MCQA zero-shot inference***

```
Question:  {question}
Options:  {options}

You should ONLY choose the letters from the options and answer in the format of 'So the answer
is X', where X is the option letters A, B, C, etc.  Please think step by step.
```

Figure 9: Zero-shot inference prompt for MCQA evaluation.