# OpenReview forum: "BOW: Reinforcement Learning for Bottlenecked Next-word Prediction"
_ICLR.cc/2026/Conference — Submitted to ICLR 2026_

### Official Review · Reviewer_DG9i · 2025-10-27

**Soundness:** 3
**Presentation:** 3
**Contribution:** 2
**Rating:** 4
**Confidence:** 4

**Summary:**

This paper introduces BOttlenecked next-Word exploration (BoW), a method to fine-tune Large Language Models (LLMs) in the Reinforcement Learning (RL) formulation.  BoW extends the standard next-word prediction (NWP) training pipeline with an intermediate reasoning bottleneck, and assigns the reward according to the likelihood of the gold next token conditioned the trajectory. This formulation naturally aligns with Reinforcement Learning (RL) and can be optimized using existing policy gradient methods such as GRPO. The authors further introduce an L1-style regularizer to avoid reward hacking. Experiments on Qwen2.5-7B-Instruct and Llama3.1-8B-Instruct demonstrate the effectiveness of propose method on several benchmarks

**Strengths:**

1. The paper is clearly written.
2. The proposed method outperforms baselines on 7 out of 10 benchmarks
3. The author does comprehensive analysis on the algorithms.

**Weaknesses:**

1. **Unclear method name**: the method "Bottlenecked next-World exploration (BOW)" in terms of both "bottleneck" and "next-world exploration".
    - As discussed is related work and Section 3.2, "bottleneck" is just a reasoning trajectory before the final prediction. Therefore, I don't see any benefit of renaming the "reasoning trajectory" into "bottleneck"
   - From my experience, "next-world prediction" refers to the pretraining task of LLM, where the loss is applied to all the tokens in the sentence. In comparison, BOW only considers the probabilty of "gold token", which is a form of outcome reward.
2. **Lack of Novelty**: based on the discussion above, BOW is a method of Reinforcement Learning with outcome reward, which has been studied on many existing works. Particularly, the hard reward (HR) formulation corresponds to standard RLVR, while the soft-reward has also been studied on papers such as [1]
3. **Training is done on Instruction-tuned model**: The experiments are conducted on instruction-tuned model, which has been fine-tuned on some of the evaluation sets. This might also partially expains that why the improvement of BOW against Vallina is not substaintial.

[1] VeriFree: Reinforcing General Reasoning without Verifiers. arXiv preprint arXiv:2505.21493

**Questions:**

1. What's the difference between BOW and RLVR (HR in Table 1) and this method [1]?
2. Why in Table 1, the performance of HR degrades compared to Vallina?
3. What would be the performance on BOW if trained on Base model?

[1] VeriFree: Reinforcing General Reasoning without Verifiers. arXiv preprint arXiv:2505.21493

---

> ### Author Response · Authors · 2025-11-18
> **Rebuttal [1/2]**
>
> We thank the reviewer for the critical feedback and address them in the following.
>
> ### **Weakness 1**
> - “Bottleneck” does not simply refer to the generated reasoning trajectory. In our case, the bottleneck learning is that, instead of directly predicting the next word, the model must first reason about probable next words. Then, the next word is recovered solely from the reasoning generated. More generally, instead of directly learning $x => y$, first learning $x => z$, then $z => y$, can be referred to as bottleneck learning. This bottleneck notion has a long history and has been widely adopted in works with related approaches that we take insights from, such as Tishby et al. (2000) and Alemi et al. (2016).
> - We are not entirely sure about what the reviewer's critique is in the second sub-comment. We would appreciate it if the reviewer could provide more clarification. However, we want to emphasize that next-word prediction does not equal pretraining; it is just a task format we use as mid- or post-training. Even though we use the gold token as the label for cross-entropy-based NWP, our motivations and setup are different. We aim to rethink the causal language modeling of the vanilla next-word prediction with RL. We treat next-word prediction as a fundamental task for the model and use pre-training corpora. For vanilla next-word prediction, models are directly optimized to predict the next word using cross-entropy loss. In our setup, the models are optimized to generate better next-word reasoning trajectories that are more accurately aligned with the next word, and we use RL to achieve this.
>
> ### **Weakness 2**
> Our novelty begins from the very start, where we motivate the entire research. We aim to rethink the causal language modeling of the vanilla next-word prediction task with RL to better incentivise the model’s general reasoning capabilities. Prior work trains RL models on supervised data, for example, math reasoning or question-answering data from General-Reasoner, as the reviewer noted. Unlike them, we treat next-word prediction as a fundamental task for the model and use pre-training corpora. Thus, just like NWP, our method does not require any annotation. With this motivation, our work shows that the model can learn fine-grained word-level reasoning from plain text rather than merely memorizing words through vanilla next-word prediction. And this learned word-level reasoning can improve the model’s generalization capability, as demonstrated by 10 out-of-distribution benchmarks we evaluate on. It is natural to understand that the goal of our work is not to pursue SOTA performance on specific downstream tasks via RL.
>
> We are not proposing a new RL algorithm. Our work is not merely "a method of RL with outcome reward". We are indeed using RL with outcome reward, but our novelty lies in the task, how we use it, and pretty much everything other than the optimization algorithm itself. And even “outcome reward” itself is a comprehensive, high-level topic, so any work can’t be deemed non-novel just because it is related. Analogously, a paper that uses SFT in a novel setup cannot be considered "not novel" just because SFT is widely used in other papers. We recognize that RL and soft rewards are not new algorithms, and we never claim they are. The novelty lies in where RL is applied and in the benefits of soft rewards for our task setting.
>
> ### **Weakness 3**
> We use an instruction-tuned model because we need it to output reasoning trajectories for next words. Thus, we expect the model to possess the basic ability to follow instructions and organize its reasoning. Regarding insignificant improvements over the vanilla model, we want to remind the reviewer that we are not using any supervised training data. We do not intend to perform SFT or RL using supervised training data, such as mathematical QA, to achieve SOTA performance, as most current works would do. Instead, we aim to explore how to improve the model’s general reasoning capability by leveraging plain text without any human supervision. It is overly stringent to undermine the contribution of our work by expecting significant improvements from a small-scale training with only next-word level weak supervision.
>
> Also, starting from instruction‑tuned models is standard practice in post‑training pipelines because it stabilizes formatting/structure and improves instruction‑following before any RL phase, as seen in InstructGPT and DeepSeek-R1. We follow this well‑established recipe for the reliability of reasoning traces: BOW is a post‑training adaptation rather than pre-training from scratch.
>
> *Tishby, N., Pereira, F. C., & Bialek, W. (2000). The information bottleneck method (arXiv preprint arXiv:physics/0004057). https://doi.org/10.48550/arXiv.physics/0004057*
> *Alemi, A. A., Fischer, I., Dillon, J. V., & Murphy, K. (2016). Deep Variational Information Bottleneck (arXiv preprint arXiv:1612.00410v7). https://doi.org/10.48550/arXiv.1612.00410*

---

> > ### Author Response · Authors · 2025-11-18
> > **Rebuttal [2/2]**
> >
> > ### **Questions**
> > 1. In natural language, the “next word” often admits multiple plausible continuations. A hard, exact‑match reward (our HR baseline) sparsifies feedback, crediting only one token and penalizing equally plausible alternatives, thereby narrowing exploration and amplifying overconfidence. Our soft reward conditions on the reasoning trajectory and assigns graded credit across the entire next‑token distribution, yielding denser, better‑calibrated updates. Regarding VeriFree, as the reviewer mentioned, the task setting is different: they use QA data obtained from human annotation or large-scale model distillation, whereas we only use plain text with next-word supervision.
> > 2. Table 1 is the zero-shot general reasoning evaluation, which is in a different format from the next-word prediction training paradigm (out-of-distribution format evaluation). We have explained the two evaluation paradigms on line 268. So, the results simply show that HR generalizes poorly.
> > 3. We have explained why we chose an instruction-tuned model in response to weakness 3, and exploring the base model is for future work.

---

### Official Review · Reviewer_LQAb · 2025-10-31

**Soundness:** 2
**Presentation:** 3
**Contribution:** 2
**Rating:** 4
**Confidence:** 4

**Summary:**

The paper proposes a reinforcement learning framework for training large language models (LLMs), called Bottlenecked Next Word Exploration (BOW). Unlike standard next-token (word) prediction, the model is prompted to generate a reasoning trajectory that analyzes the next-token prediction based on the previous tokens. A frozen scorer (also an LLM) evaluates the trajectory by computing the likelihood probability of the true next token, conditioned on both the context and the generated trajectory. The policy model (the LLM being trained) is optimized using Grouped Reward Policy Optimization (GRPO).

The authors conduct experiments on two LLMs, Qwen2.5-7B-I and LLaMA3.1-8B-I, with LLaMA3.1-8B-I also being the frozen scorer. The model is trained on narratives from the murder mystery domain, and the data are filtered to exclude context–next-word pairs where the next tokens do not require reasoning to infer. The proposed method is compared against the base LLM and several other training approaches, including selective language modeling, hard reward, and thoughts of words. The evaluation covers various general reasoning benchmarks as well as intrinsic next-word prediction tasks. Overall, the proposed method shows improvements over the original models on most tasks and outperforms other methods according to the reported results.

**Strengths:**

1. The paper generally tackles an interesting and important question about the training signals in LLMs and contributes to the line of research that explores RL as an alternative. Among the work that uses RL to post-train LLMs for incentivizing reasoning ability, the paper has some originality in applying it to a token or word level and in using a soft reward based on the reasoning trajectory rather than the final answer.

2. The paper is written with good quality and does a good job of clearly presenting what it does. The authors structure the paper well and provide detailed explanations of their design choices.

3. The experimental setting is generally valid. The compared methods and evaluated datasets seem comprehensive, and they also provide detailed ablation studies to examine effects such as reward regularization, different scorers, and the effect of training data filtering.

**Weaknesses:**

1. My main concern is the validity of the experimental comparisons. From Table 2, almost all baseline methods SLM, ToW, HR reduce the performance of the vanilla untrained LLMs on most datasets, especially for LLaMA3.1 8B I. I expect this happens because the authors rerun these methods by training models on the same, very limited murder mystery domain, which may cause the model to overfit and perform worse on other datasets. If the authors followed the original papers, the scores should not be that low. So even if the proposed method is better than other methods, the improvement could be due to less overfitting rather than enhanced reasoning ability, especially considering that the gains over the vanilla model are marginal, and in several cases, performance even drops a bit. This also leads me to question the validity of the training data. I understand the murder mystery data may demand reasoning, but there are other domains, such as math and coding. Why do the authors not train on a more diverse set of data? And for a fair comparison, why not keep the training closer to the baseline methods rather than adapting it to your data?

2. Besides, compared with a more popular RL tuning pipeline such as in the GRPO paper, this work applies RL tuning at a more fine grained token level, and the reward computation is also different. It is not clear to me from the paper why these two aspects are important. In the experiments, not all tokens are used, only selected tokens, which may be trivial, and results could vary across domains. I would be interested to see, for example, beyond the murder mystery domain, whether for math problems doing RL at the token level instead of on the final answer is better. Similarly for the reward model. I do see from the baselines that there is a hard reward comparison, but it is still not clear to me why a soft reward is better, and why not let the policy model directly output the answer after the reasoning trajectories and use that soft or hard as the reward. There are several design choices, but I am not clear why the proposed one is the best.

3. (Minor points.) From the abstract and introduction, the paper seems to frame the approach as completely shifting the supervision signal from next token prediction to an RL framework, but the experiments look more like a post training technique, since it needs LLMs trained with next token prediction as a good starting point for both the policy and the reward models. Stating this clearly could reduce confusion. Also, the paper does not discuss its limitations, and I recommend adding a section on that. Clearly stating when the method works and when it does not would bring more benefits to the community.

**Questions:**

Please refer to the weakness section.

---

> ### Author Response · Authors · 2025-11-18
> **Rebuttal [1/3]**
>
> We thank the reviewer for the critical feedback and address them in the following.
>
> ### **Weakness 1**
> First, to avoid confusion, we assume the reviewer is referring to Table 1 (general reasoning) rather than Table 2 (intrinsic next-word prediction). In Table 2, where models are trained and evaluated on the same task family (NWP), nearly all learning schemes except SLM outperform the vanilla instruction model by a clear margin across both base families. By contrast, Table 1 probes zero-shot general reasoning, which is out-of-distribution relative to all NWP-style training (including ours and the baselines). The two tables thus answer different questions: zero-shot transfer of general reasoning (Table 1) and intrinsic NWP competence (Table 2)
>
> On the training corpus (“murder mystery”):
> The term describes a story format, not a narrow topical domain. These narratives cover everyday social interactions, temporal and causal structure, spatial relations, and world knowledge (e.g., family, school, competitions, misunderstandings). In both coverage and intent, this is close in spirit to widely used book-style corpora, such as BookCorpus. Training general-purpose reasoning on such story-like plain text data is, therefore, natural rather than domain-specialized text, such as math and code.
>
> On “overfitting to mystery”: there is no overfitting happening caused by the corpus, and the evidence points to Table 2 (in-distribution NWP). In Table 2, all baselines are trained on the same corpus, and we see consistent gains for the RL-style methods and ToW. The reduced zero-shot transfer on Table 1 for SLM/ToW/HR indicates that not all learning schemes transfer equally to generic reasoning benchmarks under matched supervision. This is precisely why we also study learning-scheme generalizability: BOW transfers more robustly than these baselines. Moreover, no evidence indicates that baselines would work in our setting, which is more challenging than others and requires more robust learning schemes. Evaluating on broad suites like BBH and MMLU is the appropriate way to stress-test that generalizability.
>
> On Llama-3.1-8B-Instruct specifically: the reviewer highlighted is a model-family-specific degeneration we already analyzed in the paper. This is explained in lines 316-318 and 325-327. There is no evidence that the training data is inappropriate; it reflects how specific learning rules interact with the base model under a small-scale adaptation.
>
> On “following the original papers” and fairness of comparisons: our goal is to compare learning schemes under the same conditions, including data, supervision tokens, and decoding/evaluation, so that differences should be attributed to the training objectives, not to data scale or domain. We therefore keep as many details as possible identical across SLM, ToW, HR, and BOW. As a sanity check, our vanilla references reproduce publicly reported numbers for Llama-3.1-8B-Instruct and Qwen2.5-7B-Instruct within expected variance, which indicates the evaluation harness is fair and reproducible (See the vanilla model’s results in Table 1).

---

> ### Author Response · Authors · 2025-11-18
> **Rebuttal [2/3]**
>
> ### **Weakness 2**
> We thank the reviewer for raising questions about the design choices. We address (i) why we apply RL at the token (NWP) level in our setting, (ii) why we use a soft, distributional reward rather than a hard reward, (iii) why we select informative tokens instead of updating on every token, and (iv) the relationship to RLVR and domain considerations.
>
> (i) Why RL at the token (NWP) level in our setting: Our work targets plain text without verifiable final answers, i.e., raw next‑word supervision rather than instance‑level QA. Popular RLVR pipelines assume verifiable, instance‑level rewards (e.g., exact‑match math/code answers), which are not available in our corpus. We therefore operate at the NWP granularity, where the model must form a reasoning trace before predicting the next token. This choice is not only natural for corpora with plain text; it is also aligned with a growing body of work that explores fine‑grained (token‑level) credit assignment in RL/post‑training for LLMs (Yoon et al., 2024; Zeng et al., 2024). Our paper’s abstract already emphasizes a “brief BOW adaptation phase” rather than pretraining, and we will clarify this stage explicitly. In contrast, GRPO‑style RLVR excels when a sequence‑level, verifiable reward exists (math, code, table tasks), as shown in recent GRPO/RLVR papers and analyses; our contribution is complementary and specifically addresses token‑level supervision from raw text.
>
> (ii) Why a soft, distributional reward (vs. hard): In natural language, the “next word” often admits multiple plausible continuations. A hard, exact‑match reward (our HR baseline) sparsifies feedback, crediting only one token and penalizing equally plausible alternatives—this both narrows exploration and amplifies overconfidence. Our soft reward conditions on the reasoning trajectory and assigns graded credit across the entire next‑token distribution, yielding denser, better‑calibrated updates while discouraging “name‑the‑answer” shortcuts; this is consistent with emerging token‑level reward modeling (continuous/graded signals), which have been shown to stabilize learning relative to binary sequence‑level scores. We will tighten the motivation and pointer to our ablations in the paper. Regarding the suggestion to “let the policy output the answer after the reasoning and reward that”: in our plain-text NWP regime, there is no single verifiable ‘final answer’ per instance beyond the next token itself. Forcing a one‑token commitment collapses uncertainty and reintroduces the sparsity of exact‑match rewards. Scoring the distribution over next tokens based on the reasoning trajectory preserves uncertainty while still providing token‑level credit assignment, and it is precisely what we need.
>
> (iii) Why select (non‑trivial) tokens instead of updating on all tokens: Updating uniformly on every token dilutes the learning signal with uninformative or low‑value positions. A series of recent studies (e.g., Lin et al. (2024), mentioned as SLM in baselines) show that not all tokens contribute equally and that selective/token‑focused training improves efficiency and quality; our selection mechanism follows this established practice to concentrate RL updates on ambiguous, semantically pivotal positions. We will also add a brief clarification for this motivation in our introduction.
>
> (iv) Relationship to GRPO/RLVR and domain considerations: We borrow GRPO/RLVR's exploration spirit but target a different supervision regime: token‑level NWP over plain text. GRPO/RLVR is naturally suited to verifiable domains such as math/code; our study focuses on general LLM reasoning in narrative/book‑like text, where the reward is not verifiable at the sequence level. Evaluating whether next-token RL in plain texts vs. final‑answer RL in math is preferable in augmenting the LLM reasoning is an interesting orthogonal direction; existing GRPO/RLVR results already demonstrate strong performance for math and code, while our contributions address the complementary plain text setting, i.e., no supervision except for next tokens is provided.
>
> *Yoon, E., Yoon, H. S., Eom, S., Han, G., Nam, D. W., Jo, D., On, K.-W., Hasegawa-Johnson, M. A., Kim, S., & Yoo, C. D. (2024). TLCR: Token-Level Continuous Reward for Fine-Grained Reinforcement Learning from Human Feedback (arXiv preprint arXiv:2407.16574). https://doi.org/10.48550/arXiv.2407.16574*
> *Zeng, Y., Liu, G., Ma, W., Yang, N., Zhang, H., & Wang, J. (2024). Token-level Direct Preference Optimization (arXiv preprint arXiv:2404.11999). https://doi.org/10.48550/arXiv.2404.11999*
> *Lin, Z., Gou, Z., Gong, Y., Liu, X., Shen, Y., Xu, R., Lin, C., Yang, Y., Jiao, J., Duan, N., & Chen, W. (2024). Rho-1: Not all tokens are what you need (arXiv preprint arXiv:2404.07965 v4). https://doi.org/10.48550/arXiv.2404.07965*

---

> > ### Author Response · Authors · 2025-11-18
> > **Rebuttal [3/3]**
> >
> > ### **Weakness 3**
> > We agree that clarity on BOW's position in the training pipeline is essential. We will revise the abstract and introduction to state that BOW is a brief post‑training adaptation explicitly. Our current abstract actually already describes “a brief BOW adaptation phase”, but we will make the post‑training framing more prominent to avoid any ambiguity. In fact, NWP is just a task format, and we view it as a generic reasoning task that can be well applied in mid- and post-training stages. We are happy to include a Limitation section, as the reviewer suggested, in the final version to address this.

---

### Official Review · Reviewer_viTt · 2025-11-01

**Soundness:** 3
**Presentation:** 3
**Contribution:** 3
**Rating:** 4
**Confidence:** 3

**Summary:**

The paper proposes a new reinforcement learning framework, **Bottle-necked Next-Word Exploration (BOW)**, as an alternative to standard next-word prediction (NWP) for large language models. Instead of directly predicting the next word, BOW forces the model to first generate an explicit **reasoning trajectory**, that evaluates how well the trajectory supports the correct next token. An **L1-style reward regularizer** is introduced to discourage shortcut behaviors like “naming the answer” and to encourage more general reasoning.

**Strengths:**

1. **Comprehensive Evaluation and Analysis** – The paper includes detailed ablations (scorer choice, regularization, data filtering) and human studies, demonstrating robustness of the method’s behavior.

2. **Novel Soft Reward** – The soft, probabilistic reward may offer smoother and denser feedback than hard binary rewards, improving exploration efficiency and stability during training.

3. **Empirical Performance Gains** – BOW outperforms strong baselines (RPT, ToW, SLM) across several reasoning benchmarks and majority of intrinsic NWP evaluations.

**Weaknesses:**

1. **Lack of Cost Analysis** – This is a big concern. Generating reasoning trajectories for next token prediction is computationally expensive; the paper does not quantify training time, GPU cost, or sample efficiency relative to simpler continual-pretraining methods.

2. **Narrow Experimental Domain** – Training data seem to come from *murder-mystery narratives*, a very specific genre. This raises the question of how the proposed model performs well on generic reasoning tasks, such as coding, mathematics, or dialogue. It would be useful to have a detailed description of training stages. For each stage, we need to compare the training cost and the training domain used side by side with the simple autoregressive baselines or SLMs for clarity.

**Questions:**

1. How costly is the proposed approach compared to vanilla next token prediction baseline?
2. How does the performance of the proposed approach generalize to out-of-domain reasoning tasks?

---

> ### Author Response · Authors · 2025-11-18
> **Rebuttal [1/2]**
>
> We thank the reviewer for the critical feedback and address them in the following.
>
> ### **Weakness 1**
> While we recognize that training cost is essential for both scientific evaluation and practice, a comprehensive cost engineering study is outside the scope of this work. Our goal is to investigate whether explicit next-word reasoning signals learned with RL improve generalizations in reasoning, not to optimize wall-clock efficiency. That said, the broader literature supports the premise that additional optimization or computation can be warranted when studying generalization. Many RL papers, such as DQN (Mnih et al., 2013) and A3C (Mnih et al., 2016), have shown that more computation is needed to understand model generalization, and Power et al. (2022) show that more computation could be a primary reason models learn better.
>
> However, to benefit future works, we will report a detailed computational cost analysis. For a quick reference, we trained on 4x NVIDIA H200 GPUs as described in Appendix B.3.
> - BOW (ours): ~11h for ~40 RL steps
> - HR: ~7h
> - ToW: ~2h model training, plus ~4h for batch annotation (distillation)
> - SLM: ~1h (pure NWP continual pretraining)
>
> These times reflect the inherent RL overhead (rollout sampling + scorer inference) relative to simpler continual-pretraining baselines. For example, BOW's extra 4 hours compared to HR are due to the scorer inference process. Note that BOW’s efficiency can be improved through engineering choices, such as asynchronous reward evaluation and speculative decoding for trajectories. A thorough exploration of these system choices is valuable, but beyond our present scope.
>
> *Power, A., Burda, Y., Edwards, H., Babuschkin, I., & Misra, V. (2022). Grokking: Generalization Beyond Overfitting on Small Algorithmic Datasets (arXiv preprint arXiv:2201.02177). https://doi.org/10.48550/arXiv.2201.02177*
> *Mnih, V., Kavukcuoglu, K., Silver, D., Graves, A., Antonoglou, I., Wierstra, D., & Riedmiller, M. (2013). Playing Atari with deep reinforcement learning (arXiv preprint arXiv:1312.5602). https://doi.org/10.48550/arXiv.1312.5602*
> *Mnih, V., Puigdomènech Badia, A., Mirza, M., Graves, A., Lillicrap, T. P., Harley, T., Silver, D., & Kavukcuoglu, K. (2016). Asynchronous methods for deep reinforcement learning (arXiv preprint arXiv:1602.01783v2). https://doi.org/10.48550/arXiv.1602.01783*

---

> > ### Author Response · Authors · 2025-11-18
> > **Rebuttal [2/2]**
> >
> > ### **Weakness 2 - part 1**
> > We recognize that domain choice matters. In our paper, we use story-like narratives from a murder-mystery corpus because such stories encode who did what, when, why, and what happened next, naturally exercising commonsense and counterfactual reasoning. The chosen narratives contain many next-word pairs that demand reasoning about everyday situations rather than a single topical niche; our “Training Data” section explains this choice and the pipeline. Narrative fiction of this kind is similar in spirit to widely used book corpora (e.g., BookCorpus), which serve as standard pretraining sources precisely because they span everyday events, social interactions, and high-level semantics beyond any one domain. Given this story-like nature, it is more appropriate to evaluate general-purpose reasoning than to treat the setting as domain-specific.
> >
> > Accordingly, our evaluation covers generic reasoning benchmarks across commonsense reasoning, multi-hop reasoning, and broad knowledge: CommonsenseQA, PIQA, StrategyQA, ARC-Challenge, WinoGrande, BBH, MMLU, MMLU-Pro, and GPQA (see the Evaluation in Section 4). These suites are expressly designed to probe reasoning and knowledge transfer across many everyday and academic domains, not a single niche. It makes more sense to evaluate on the datasets we used rather than math- or code-specific ones.
> >
> > We agree that coding and mathematics are essential for LLM reasoning. Still, AI reasoning is far more than coding and math, and we view them as specialized reasoning domains that typically benefit from targeted training, optimization, and data (e.g., code- or math-specific models). People have focused more on them in recent reasoning work because of the success of RLVR, but that doesn't mean other, more generic reasoning isn't essential. In fact, generic reasoning has a more extended history. It is equally challenging, and that's precisely the benchmarks we used. Specialized math/code reasoning is orthogonal to our research focus and usually handled by domain-specific processing, as exemplified by Codex (code) and Qwen-Math (math).
> >
> > In short, our training corpus is a story format that exercises diverse everyday reasoning, and our results are validated on standard, generic reasoning benchmarks. The absence of math- or code-specific tests does not detract from our claims about general LLM reasoning; instead, those domains are distinct specialties best addressed in follow-up work with domain-appropriate training.
> >
> > ### **Weakness 2 - part 2**
> > Thank you for this suggestion again. To reiterate, our method has a single training stage: a brief BOW RL adaptation in which the policy first generates a next-word reasoning trajectory and is then optimized with a soft, distributional reward from a frozen scorer (see Section 3). There is no multi-stage pipeline beyond this RL adaptation. For domain comparability, all methods are trained/evaluated under the same corpus and matched supervision tokens (See Section 4 for the narrative corpus and Section 4’s Baselines for how the baselines use the same supervision tokens for fairness). While a complete cost-engineering study is outside our scope, we will add a table in the next version that lists, side-by-side, for BOW, HR, ToW, SLM, and Vanilla: (i) objective (RL vs. autoregressive), (ii) training domain (shared), (iii) hardware (same), and (iv) measured GPU-hours (as report in response to first weakness).
> >
> > ### **Questions**
> > 1. Please refer to the response to Weakness 1.
> > 2. Based on the pre-training corpus, the benchmarks we evaluate on are already out-of-domain tasks that are suitable under our setting and experiment scale.

---

### Official Review · Reviewer_q7pv · 2025-11-02

**Soundness:** 3
**Presentation:** 3
**Contribution:** 3
**Rating:** 6
**Confidence:** 3

**Summary:**

This paper reformulates the next-word prediction in LLMs as a reinforcement learning problem, instructing the policy model to output a reasoning chain that carries critical information about the gold next word information. Rewards are assigned from a separate frozen LLM scorer, which predicts the next word based on the given reasoning chain, outputting a soft probability score. The method, named bottlenecked next word exploration (BOW), achieves encouraging reasoning performance on several benchmarks and models.

**Strengths:**

* The idea of converting next-word prediction into a reinforcement learning issue is interesting, which also provides a way to scale RL up.
* The paper is generally well written and easy to understand.
* Experiments are based on several models and benchmarks, making the results more convincing.

**Weaknesses:**

* It’s unclear how BOW scales as training data increases: the experiments are only based on one data setup.
* The hard reward baseline performs much worse than the vanilla baseline, which may be misleading.

**Questions:**

In Dong et al. (2025)’s work, the hard reward also delivers encouraging performance, but in the experiments, this method achieves much worse performance even than the vanilla baseline. Why would this happen? Is this caused by sub-optimal optimization?

Another way to understand the effectiveness of hard reward is to convert the soft score in BOW to binary, such as if the score is larger than some threshold (like 0.2), the model gets a positive reward of 1, otherwise 0. This would better illustrate how hard and soft reward works.

---

> ### Author Response · Authors · 2025-11-18
> **Rebuttal**
>
> We thank the reviewer for the critical feedback and address them in the following.
>
> ### **Weakness 1**
> We are not aiming to achieve SOTA performance on downstream tasks, and we are not scaling experiments at the pre-training level; instead, we explicitly mention “a brief BOW adaptation phase” in the abstract. The research goal of our work is to explore teaching models with more fine-grained and better next-word reasoning, using only plain text. Our experiments are trained on \~45k tokens (\~300k raw tokens before filtering), which is sufficient to provide scientific value. Such a scale also offers a unique perspective on the effectiveness of various learning objectives from prior related work (the baselines we use), demonstrating that our proposed method is more robust to overfitting and generalizes better. Whether BOW is a complete substitute for vanilla NWP is for future work.
>
> ### **Weakness 2**
> We appreciate the concern. We believe this weakness relates to the question in the next section, so we address it together here. Our HR is a faithful reflection of performing the reward design method proposed in Dong et al. (2025) within our setting and scale. The purpose of HR in our study is to isolate reward design effects under the same data and base models, not to re-implement a different pretraining pipeline. As analyzed in our paper (both qualitatively and quantitatively), a hard, correctness-only reward design suffers from overfitting and overconfidence (e.g., more hallucinations on TruthfulQA). In contrast, our soft, distributional signal supplies dense partial credit and supports better exploration. In our controlled comparisons, this yields the expected pattern: HR lags, while our soft-reward method BOW improves and generalizes more reliably.
>
> Crucially, comparing HR to Dong et al. (RPT) is not apples-to-apples and does not contradict our findings. RPT starts from a much stronger base model (R1-distill-Qwen-14B) and performs “reinforcement pre-training” on a specialized math corpus (OmniMATH). In contrast, we intentionally study reward design on a more general reasoning corpus with instruction-tuned models that exhibit fewer biases in reasoning. RPT’s reported gains, therefore, combine stronger initialization and domain-specific training; this conflation weakens their own claims in “reinforcement pre-training”. Even the RPT paper itself notes that its experiments primarily use a 14B model and that the (continual) pretraining corpus is predominantly mathematical and knowledge-driven (only two benchmarks), underscoring the limited breadth of evidence for general-domain reasoning transfer. Given all the differences, there is no sign that the RPT-like method will perform better than what we are reporting in our setting. In our setup, our empirical results and findings are consistent and diagnostic rather than misleading: Table 1 shows that HR commonly underperforms vanilla instruction models on general reasoning. BOW delivers the strongest and most stable improvements across benchmarks. Table 2 also shows that our training is not overfitting, as the HR baseline could perform better than the vanilla model.
>
> ### **Question**
> Please refer to our response to the second weakness, and we appreciate the reviewer's one alternative variant of the hard-reward design.
>
>
> Dong, Q., Dong, L., Tang, Y., Ye, T., Sun, Y., Sui, Z., & Wei, F. (2025). Reinforcement pre-training (arXiv preprint arXiv:2506.08007 v1). https://doi.org/10.48550/arXiv.2506.08007

---

### Author Response · Authors · 2025-11-25
**Summary of Rebuttal & Invitation for Follow-up**

Dear Reviewers and Area Chairs,

Thank you for your time and thoughtful feedback on our submission. We have posted point-by-point responses under each review and would be grateful if you could take a look and continue the dialogue while the discussion window is open. We have also provided a summary of our paper and the rebuttal highlights for AC’s reference.

## What BOW contributes
BOW reframes next‑word prediction as RL with an explicit reasoning bottleneck: the policy first produces a trajectory, which is scored by a frozen evaluator using a soft, distributional reward equal to the gold next‑token probability conditioned only on that trajectory; we also include an optional L1‑style term to discourage shortcuts. On two base models (Qwen2.5‑7B‑Instruct, Llama3.1‑8B‑Instruct), a brief BOW adaptation 1) improves general zero‑shot reasoning across 10 benchmarks, outperforming strong continual‑pretraining and RL/SFT alternatives by ~5% on average, 2) achieves the best performance among 7 out of 10 intrinsic NWP evaluations compared with baselines, and 3) leads to the most human preferred next-word reasoning trajectories compared with those generated by baselines.

## Rebuttal highlights
- Our methodology vs. contemporary RL: We clarified how our bottlenecked next-word exploration with soft rewards differs from hard/binary rewards, and why it puts greater pressure on explicit intermediate reasoning rather than surface fluency.
- Our training corpus is general: It resembles book corpora (e.g., BookCorpus), a widely used standard pre-training source that covers everyday events, social interactions, and high-level semantics beyond a single niche domain.
- Empirical evaluation is broad, not narrow: A brief BOW adaptation on Qwen 2.5-7B-Instruct and Llama3.1-8B-Instruc improves zero-shot reasoning across 10 benchmarks, with ~5% average improvement and top results on 7/10 intrinsic NWP evaluations.
- Justification for the absence of math: Math-related training and evaluation are orthogonal to our current work. The results across 10 evaluation benchmarks, which span a wide range of general reasoning capabilities, are sufficient to warrant our scientific contribution.
- Comparisons are solid and aligned: We benchmarked against competitive continual-pretraining baselines, including a contemporary RL variant with a hard reward and an SFT approach with augmented data; BOW shows consistent average gains under the discussed settings.

## Kindly request follow-ups and updated evaluations
If our rebuttal resolves your main concerns (e.g., novelty v.s. prior RL, fairness of comparisons, or robustness), we would appreciate an updated review and score to reflect the new evidence or more follow-up discussions. We remain fully available to clarify more details in the remaining days before the discussion window closes. Thank you again for your engagement and for helping us improve the paper. All discussions during the discussion phase will be updated into the camera-ready version of the paper.

Best regards,
Authors

---

### Meta-Review · Area_Chair_814C · 2025-12-18

**Summary:**

As mentioned by the authors, the primary objective of this work is to explore how to improve the model’s general reasoning capability by leveraging plain text without any human supervision. However, I concur with Reviewers viTt and LQAb that relying exclusively on murder mystery datasets is insufficient to substantiate claims regarding "general reasoning capability." To convincingly support such a broad assertion, the evaluation must incorporate a more diverse array of datasets and tasks.

Furthermore, while the authors provided a high-level overview of the proposed method's cost breakdown, a comprehensive comparative analysis against existing baselines is needed. Such a comparison is necessary to properly justify the method's practical utility as generating reasoning trajectories is costly.

**Reviewer Concerns:**

The authors have (partially) addressed the following concerns:
- Misleading hard reward baseline. [q7pv]
- Why is doing RL at token level superior and would it generalize to reasoning tasks at other domains? [LQAb]
- Why is the proposed soft reward model better than the hard one? [LQAb]
- Writing issues, e.g., confusing position of the proposed framework and missing discussion on limitations. [LQAb]
- Unclear naming of the method. [DG9i]
- Lack of novelty. [DG9i]

However, the following concerns are still outstanding:
- Unclear how BOW scales as training data increases. [q7pv]
- Concern on the validity of the experimental comparisons. The improvement could be due to less overfitting rather than enhanced reasoning ability, especially considering that the gains over the vanilla model are marginal. [LQAb]
- Limited data domain (only murder mystery data). [viTt, LQAb]
- Lack of cost analysis. [viTt]
- Training on instruction-tuned models causes marginal benefits. [DG9i]

**Reviewer Scores:**

For concerns that require clarification, fully participating in the discussion would possibly lead Reviewers DG9i and viTt to consider changing the scores.
But for the major concern on the limited data domain, as the authors did not provide any experimental evidence, it is hard for Reviewers viTt and LQAb to change their mind.

---

### Decision · Program_Chairs · 2026-01-26

Reject